# Effort and time costs influence motivational asymmetries in self-benefitting vs pro-environmental decisions
Boryana Todorova [1], Lei Zhang [1,2,3,4], Lukas Lengersdorff[1], Kimberly C. Doell [1,5,6], Jonas P. Nitschke [1], Paul A. G. Forbes[1,7], Sabine Pahl [1,6] & Claus Lamm [1,6] ✉

Mitigating climate change requires individuals to adopt more pro-environmental behaviours, many of which come at a personal cost. Costs such as the time and effort associated with certain behaviours are integral to everyday decision-making and can significantly shape people's motivation to act. In this preregistered study, we employed an experimental paradigm designed to quantify how people discount effort (measured via a grip-force device) and time (operationalised as waiting time) for self-benefitting and pro-environmental outcomes. Participants ($n = 74$) could earn monetary rewards for themselves (in half of the trials) and for reducing carbon emissions (in the other half). We observed a higher willingness to incur time and effort costs for self-benefitting than for pro-environmental outcomes, in particular when the rewards offered were higher. Moreover, computational modelling revealed rewards were discounted nonlinearly by both time and effort: effort discounting was best described by a parabolic function, and temporal discounting by a hyperbolic function. Finally, when linking experimental behaviour to self-report measures, we found that participants who were more motivated to invest time and effort for the environment also reported greater willingness to support costly climate change mitigation policies, whereas climate change beliefs were not significantly associated with the cost-incurring task behaviour. Our approach highlights differences in how individuals respond to costs associated with personal vs environmental benefits and presents a promising tool for further research on environmental decision-making.

Climate change is one of the greatest threats we face today. With estimates suggesting that 85% of the global population is already impacted by climate change[1], the need for action is evident. Rapid and significant emissions reductions require swift and decisive individual actions alongside systemic changes. Household consumption alone accounts for 72% of global greenhouse gas emissions[2], and numerous daily choices—related to mobility, consumption, housing, and nutrition—play an important part in emission trajectories. These choices, however, are often costly to the individual. While financial costs, such as higher prices for sustainable products

or transportation, are commonly recognized as relevant in this respect[3–5], psychological costs can also play a critical role. Actions such as choosing a longer train ride over a shorter flight, engaging in climate activism, or recycling, inherently also carry psychological, non-financial costs, such as the effort and time people put into them. This is likely to impact the motivation to engage in such pro-environmental actions.

Motivation is a crucial component of decision-making and often involves weighing the expected benefits of an action against the associated costs[6]. Effort discounting and temporal discounting are phenomena that

[1]Department of Cognition, Emotion, and Methods in Psychology, Faculty of Psychology, University of Vienna, Vienna, Austria. [2]Centre for Human Brain Health, School of Psychology, University of Birmingham, Birmingham, UK. [3]Institute for Mental Health, School of Psychology, University of Birmingham, Birmingham, UK. [4]Centre for Developmental Science, School of Psychology, University of Birmingham, Birmingham, UK. [5]Centre for the Advanced Study of Collective Behaviour, University of Konstanz, Konstanz, Germany. [6]Environment and Climate Research Hub, University of Vienna, Vienna, Austria. [7]Institute of Experimental Psychology, Heinrich Heine University, Düsseldorf, Germany. ✉e-mail: claus.lamm@univie.ac.at

affect motivation, referring to the fact that rewarding outcomes are devalued based on the amount of effort or time associated with obtaining them. Research on motivation in humans and non-human animals has often used experimental approaches incorporating real costs and actual rewards and has shown that higher effort and longer delays are usually associated with lower perceived reward value, resulting in a lower motivation to perform the behaviour[6–13]. Notably, computational modelling has revealed that effort and temporal discounting follow different discounting shapes, suggesting that they influence decision-making in distinct ways[14]. Building on this, research on prosocial motivation has applied similar cost-benefit approaches and has shown that people exhibit lower motivation to engage in actions that benefit others, compared to actions that benefit themselves, when costs are involved[15–18]. Moreover, even when individuals choose to engage in effortful prosocial behaviour, they tend to exert less physical effort when acting for others compared to when acting for themselves, a finding described as "superficial prosociality"[16,17,19,20].

The processes of time and effort discounting are likely also integral in pro-environmental decision-making. Choosing the train over a short-haul flight, for example, demands extra travel time and inconvenience, but offers the "reward" of markedly lower emissions. However, once the perceived cost outweighs the subjective value of the reward, for instance, due to a train journey that is too long or uncomfortable, motivation is likely to decline. Thus, understanding how effort and temporal discounting shape this cost-benefit evaluation can be valuable for designing interventions and policies that boost the perceived value of these actions and/or reduce their subjective costs, thereby supporting sustained engagement in climate action.

While significant strides have been made in understanding how effort and time discounting affect motivation in many domains, this insight has not yet been fully translated into the methods used to study pro-environmental behaviour. Most commonly, pro-environmental motivation has been measured by self-report measures alone, which have several limitations, including social desirability bias, poor recollection of past behaviours, reporting bias, and consistency bias[21–23]. Moreover, they do not offer a mechanistic explanation of the processes driving motivated behaviour[6]. Thus, there is a clear need to expand the range of approaches and develop consequential behavioural paradigms to study pro-environmental behaviour that highlight cost-benefit trade-offs, allow for more experimental control, and offer opportunities to investigate different aspects of the decision-making process[4,24–26]. Encouragingly, a growing number of studies have begun addressing this gap by employing experimental paradigms that incorporate real costs and benefits[4,25,27–29]. For instance, a study focusing on cognitive effort showed that individuals put less effort into gaining monetary benefits for a pro-environmental cause than for a cause that benefitted themselves[28]. However, there is a notable gap in research that systematically compares different types of costs, such as time and effort, and that varies the magnitude of both the costs and potential rewards in order to model how they influence individual decision-making.

To address this, we conducted a preregistered experiment involving actual cost-incurring behaviours (physical effort and waiting times) with tangible outcomes: monetary rewards for the participant or for a $CO_2$ reduction program. By operationalizing pro-environmental motivation as the willingness to accept measurable personal costs (effort or time) to generate a concrete environmental benefit ($CO_2$ reduction), our paradigm moves beyond hypothetical scenarios or self-report measures to capture behavioural trade-offs in a controlled setting. Participants ($n = 74$) engaged in two tasks: an effort discounting task and a temporal discounting task. In both tasks, in half of the trials, participants could win money for themselves ("self" condition), whereas in the other half, they could win a monetary reward that would be donated to a $CO_2$ reduction program ("environment" condition). Crucially, to obtain the reward, participants had to invest 1) a certain percentage of their maximum force in the effort task (see Figs. 1A) or 2) endure a short waiting period in the time task (see Fig. 1B). The within subject design of the tasks allow us to compare pro-environmental motivation with self-benefiting motivation within individuals, helping us determine whether differences in pro-environmental motivation are specific

to the pro-environmental context and not the result of general apathy and lack of motivation. Moreover, computational modelling allowed us to investigate the potentially distinct mechanisms underlying how the perceived value of the rewards is discounted by effort and time at the trial-by-trial level.

Notably, people differ not only in how their decisions are affected by rewards and costs, but also in their underlying beliefs about climate change, and this might shape their willingness to act for the environment[30]. For example, individuals who strongly believe that climate change is real, human-caused, and harmful may be more inclined to accept personal costs to support climate change mitigation. However, there is considerable debate around whether and when beliefs translate into action in the field of environmental psychology[31–34]. Thus, we investigated whether individual differences in climate change beliefs are associated with how participants discount time and effort for environmental vs self-benefiting rewards.

Moreover, for the required large-scale societal transformation, individual actions must be accompanied by system-level action in the form of large-scale policies. Policies, in turn, are driven by individuals demanding and supporting them[35]. We thus also investigated how inter-individual differences in costly decision-making are related to support for costly climate policies. Given that policy support measures tend to be more valid when they emphasize trade-offs[26], we developed a scale for measuring climate policy support, which highlights the cost associated with each policy.

Our key preregistered hypotheses were threefold. First, we expected that participants would be more likely to select the costly option for themselves compared to the environment in both tasks. In line with research on prosociality, for the effort task, we further hypothesized that when participants chose the effortful option for the environment, they would energize their actions to a lesser extent than when exerting effort for themselves, as reflected in pressing with less force on the grip-force device, indicating a discrepancy between decisions and actual behaviour. Second, we anticipated that this reduced willingness to select the costly option for the environment in both tasks would be captured in the computational modelling as stronger discounting for environmental relative to self-benefits. Finally, we expected that the difference in discounting scores for self vs environmental outcomes would be associated with participants' climate change beliefs and their willingness to support costly climate policies.

## Methods
### Participants
We collected data from 74 neurotypical, German-speaking young adults (35 women, 37 men, and 2 other/prefer not to say; age mean = 24.7 years, SD = 3.88, range = 18–34). The targeted sample size was based on our preregistered power analysis (https://osf.io/bksf9, preregistration date: 16.03.2022), conducted for a repeated measures ANOVA model using G*Power 3.1. The preregistered plan was to use linear mixed models (LMMs); however, the sample size estimate was based on the use of repeated measures ANOVAs, which served as a fallback in case of convergence issues of the LMMs. Despite the discrepancy, given that repeated measures ANOVAs have lower power than LMMs (which account better for within-participant differences), the estimated sample size from the power analyses could serve as a conservative estimate for the targeted LMMs. Due to the lack of previous studies comparing self to environmental motivation, we assumed a medium effect size for the power analysis (Cohen's $f = 0.5$). Using a standard error probability of α = 0.05, a power of 1-ß = 0.8, a conservative ε = 0.0068 (i.e., 1/150 trials) for non-sphericity corrections, and conservative estimates of a correlation of $r = 0.5$ for repeated measurements, we estimated a targeted sample size of N = 64 participants. We then opted for a slightly larger sample of N = 74 to increase the power of detecting an effect. We compensated participants with 20 Euros for their participation, plus the bonus sum they won for themselves. No deception was used, and no data on race or ethnicity was collected. All of the participants provided informed consent, and the local ethics committee of the University of Vienna [EK01093] approved all procedures beforehand. Information on participant recruitment is included in SI, Section 2.1.

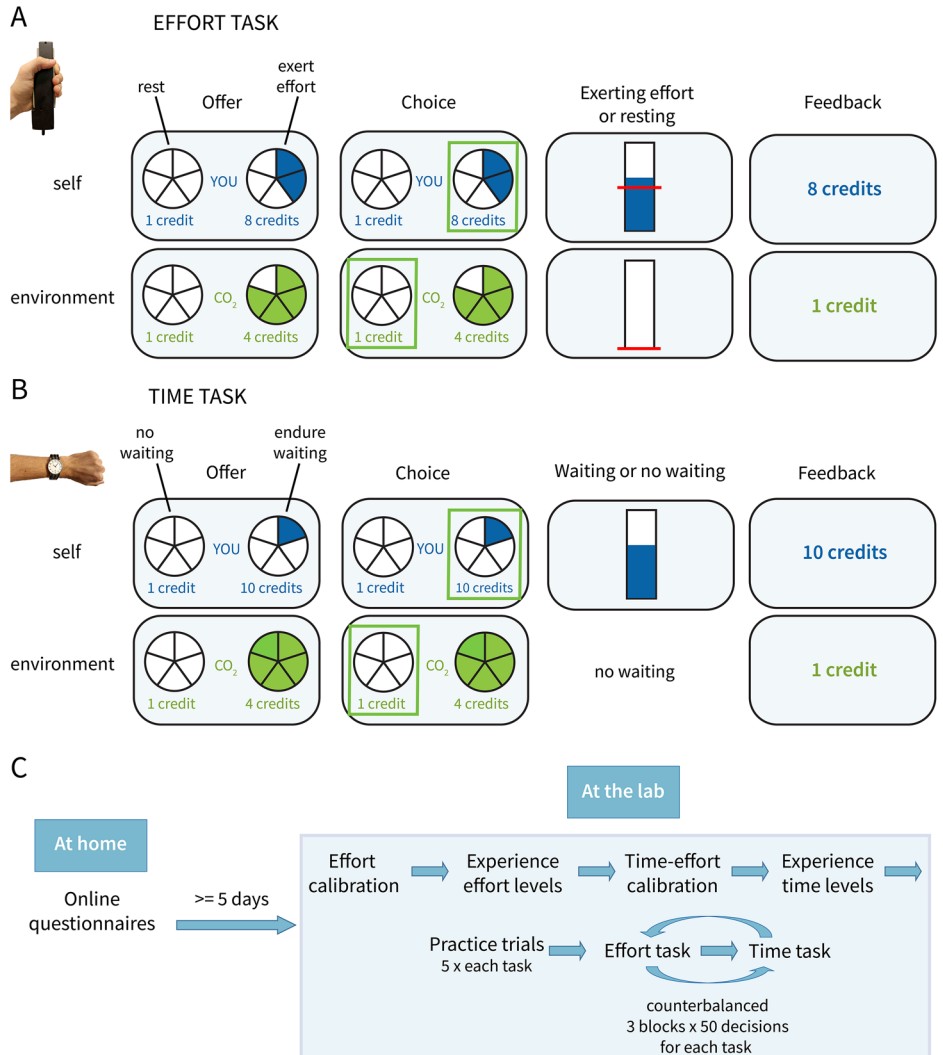

**Fig. 1 | Experimental design. A** Design of the effort task. On each trial, participants were given a choice between a no-effort option for a small reward (1 credit guaranteed) and an effort option (varying; 40, 50, 60, 70, 80% of their individual maximum; individually calibrated) for a larger reward (varying; 2, 4, 6, 8, 10 credits), with effort and reward levels designed to vary independently. If participants selected the effortful option, they were asked to exert the required force for at least 1 s within a 3-second window to earn the reward. In self-trials, participants received the reward; in environmental trials, the reward was used to reduce emissions via donations to $CO_2$ reduction program. **B** Design of the time task. The task followed the same design as the effort task, with the only difference being the cost. Here, when participants selected the costly option, they had to endure a short waiting period (individually calibrated, range 3 to 27 s). **C** Graphical overview of the experimental procedure.

## Questionnaires

Prior to coming to the lab (at least 5 days to minimize priming effects), participants completed a set of online questionnaires:

**Climate change key beliefs**. To measure participants' beliefs about climate change, we used a questionnaire[36] measuring five aspects of participants' beliefs: belief certainty (1 item), human causation (1 item), collective efficacy (1 item), harm timing (2 items), and harm extent (7 items). The exact items used are included in the SI, Section 2.3.1.

**Policy support measure**. Policy support is frequently assessed in ways that fail to reflect the real-world trade-offs that make such policies controversial or difficult to implement. To address this limitation, we developed a scale designed to highlight the trade-offs by incorporating the potential costs of the policies. We followed a structured approach in which each item presented a policy with both a widely perceived pro-environmental benefit and a commonly perceived personal or societal cost. For example: "We expand protected forest areas, even if it means

having less land for agriculture" (see SI Table S1 for the items and SI Fig. 3B for distribution of the responses). The scale contained 12 items with answer options on a Likert scale from 1 (strongly against) to 5 (strongly for). The items emphasize the cost that comes with these policies so that we can investigate whether people who are willing to invest their effort and time in the task are also more willing to support such policies when made aware of the cost.

**Other questionnaire measures**. Additionally, we administered the German versions of the following questionnaires: Egoistic, altruistic, biospheric, and hedonistic value orientation[37], Apathy Motivation Index[38] (AMI), and 29 selected items of the general ecological behaviour scale[39] (GEB). Note that the psychometric properties of the GEB allow it to be shortened in any given way. To select the items, we had three independent raters select the items they perceived as relevant for the respective target sample (young adults). We also gathered information on socio-demographic variables, including age, gender, education level, political orientation, subjective financial situation, and current occupation.

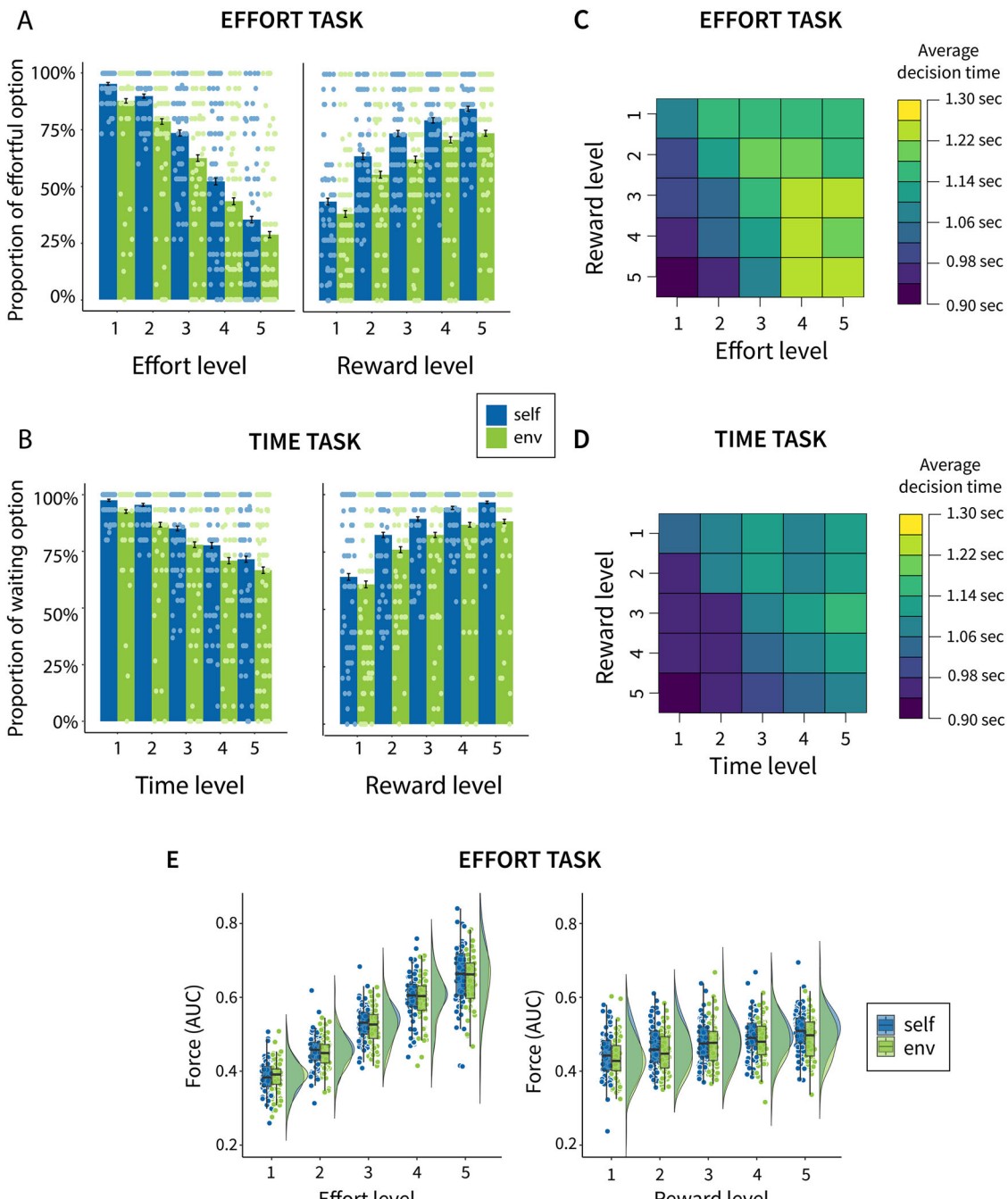

**Fig. 2 | Behavioural results (*n* = 74). A, B** Proportion of costly choices plotted for each cost and reward level and split by recipient (self and environment), (**A**) effort task, and (**B**) time task, with dots representing each individual's value and error bars indicating standard error of the mean. **C, D** Average decision time for the (**C**) effort task and (**D**) time task, plotted for each cost and reward level combination. **E** Average force expended in the effort task (calculated as area under the curve)

during the 3-second window, plotted separately for each effort and reward level and for self and environmental trials. The boxplots show the median, the first and third quartiles (bounds), and the whiskers extend to 1.5x the interquartile ranges from the bounds. The dots represent individual scores created by averaging the AUC for all the trials where an individual chooses to put effort.

## Lab procedure

Seventy-four participants completed the experimental part of the study in a single laboratory session that consisted of several stages. Upon arrival, they were welcomed to the lab, briefed, and received instructions. The session began with preparatory steps, including calibration procedures and familiarization with the effort and time levels, as well as several practice trials to ensure that participants understood the task procedure. The experimental tasks followed, where participants made repeated decisions involving effort and waiting costs for self-benefiting and pro-environmental outcomes. An

overview of the procedure is shown in Fig. 1, and each stage is described in detail below.

**Measurement of individual maximum force for effort calibration.** To ensure that the effort levels used in the task are relative to participants' individual strength, we measured individual maximum force using a maximum voluntary contraction (MVC) task, following a procedure established in previous research[17,19]. Participants were asked to grip a handheld dynamometer with their dominant hand with as much force as

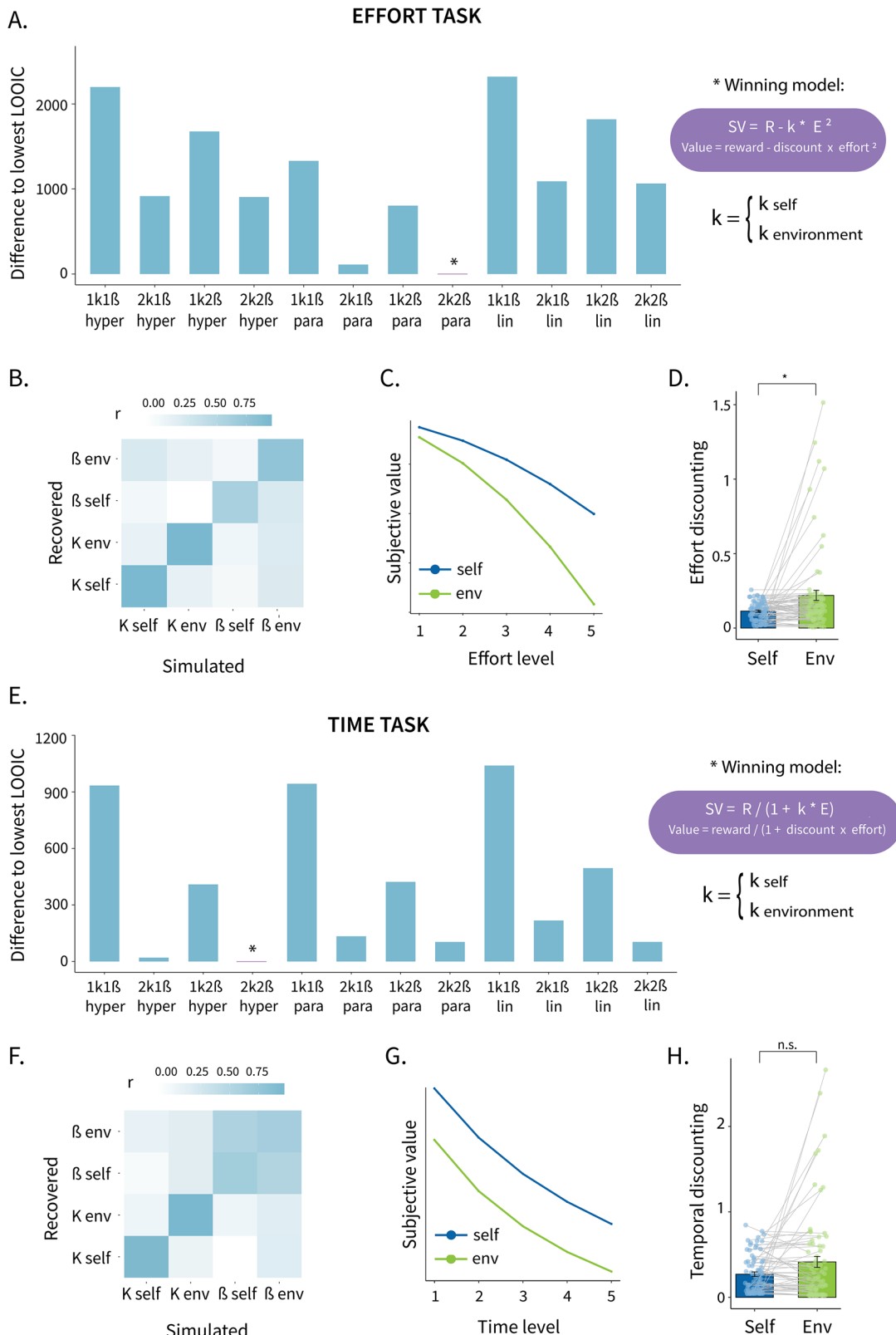

**Fig. 3 | Computational modelling results. A, E** Model fit assessed with LOOIC values, comparing the computational models of (**A**) effort discounting and (**E**) temporal discounting. Lower values indicate higher predictive accuracy.
**B, F** Parameter recovery of subject-level discounting (k self and k env) and ß parameters (ß self and ß env) for the (**B**) effort task and (**F**) time task. The correlation matrix shows the correlation between the recovered and simulated data from the winning model. **C, G** Illustrative plots indicating how the subjective value (plotted for reward level 5 for illustrative purposes) changes as a function of (**C**) effort and (**G**) time. **D, H** Individual discounting parameters for self and environmental trials (*n* = 74 participants) (**D**) for effort and (**H**) for time with statistical significance based on Wilcoxon signed-rank test, with dots representing each individual's value and error bars indicating standard error of the mean.

possible three times in a row for three seconds, with small breaks in between. The highest of the three values served as a measure of an individual's MVC.

**Familiarization with the different levels of effort**. Participants experienced each effort level used later in the task (40, 50, 60, 70, and 80% of their MVC) to familiarize themselves with how hard they had to press to surpass the respective force thresholds. The effort levels were visually indicated as pie charts with 1-5 filled segments, corresponding to the five levels of effort. Participants were sequentially presented with each pie chart level and asked to press hard enough to reach the required threshold indicated by a line on the screen, allowing them to gain a feeling for the relative difficulty of each level. Importantly, they were not informed about the exact percentage of their own maximum voluntary contraction that each level represented. Additionally, after experiencing each effort level, they were asked to complete three items about their subjective experiences of the effort levels (e.g., how physically demanding it was; see SI, Section 3.6).

**Time-effort calibration**. To ensure that physical effort and waiting time were perceived as equally costly across levels and that levels of the effort cost were matched to levels of the time cost, each participant completed a calibration procedure. On each trial of the calibration, participants were asked whether they wanted to invest their physical effort (corresponding to the lowest [40%] and highest [80%] effort levels) or wait a certain amount of time (ranging between 3 and 27 s, see SI, Fig. S1. After finding a point of indifference for the lowest and highest levels, we calculated three equidistant values between the two we calibrated, resulting in 5 levels of waiting time. More details on the calibration procedure are available in the SI section 2.2.

**Familiarization with the different levels of waiting times**. Next, participants experienced the individually calibrated five levels of waiting time. As with the effort, the waiting time levels were indicated as pieces of a pie chart and not by stating the number of seconds they would have to wait. After experiencing each time level, participants answered a few short questions about their subjective experiences of the time levels (e.g., how long they perceived the time to be; see SI, Section 3.6).

**Main experiment**

Pro-environmental effort task. The effort task is adapted from prior work[16], and was initially developed to study prosocial decision-making. On each trial, participants had to make a choice between a no effort (doing nothing) and low reward offer (always getting 1 credit) and a higher effort and higher reward offer (varying; 2, 4, 6, 8, or 10 credits), where they had to exert effort (vaying; 40, 50, 60, 70 or 80% of their individually calibrated maximum force) to win the higher reward (see Fig. 1A and SI, Fig. S2A for a decision tree). The size of the reward and the required effort were systematically varied to be independent of each other. In half of the trials, participants had the chance to win a monetary reward for themselves, and in the other half, a monetary reward would be donated to a $CO_2$ reduction program. The trial order was pseudorandomized using mini-blocks of five trials, maintaining an average effort level of 2–3 over the previous five trials to avoid fatigue. The pseudorandomization also ensured that the same recipient did not appear more than four times in a row (i.e., participants were never exposed to five or more self-trials in a row). Participants were informed that every point they win in the $CO_2$ trials would be translated into money that would be donated to an organization that actively combats climate change by reducing greenhouse gas emissions through projects that promote renewable energy sources, energy-efficient technologies, and sustainable solutions. The framing ensured participants were clearly informed that their choices represented contributions to greenhouse gas reductions, rather than broader prosocial causes (see SI, Section 2.5, for exact wording of the instructions). Participants had 3 seconds to make a decision, and when they selected the effortful option, they had 3 seconds to exert the required level of effort. They had to attain the required effort level for (at least) 1 second during this 3-second window to attain the reward, which ensured that participants had a fair opportunity to succeed while also maintaining meaningful consequences for not reaching the target. Visual real-time feedback was provided to show participants how much effort they were exerting in comparison to the requirement (indicated by a red line, see Fig. 1). After that phase, participants received feedback on how many credits they won. If the participant chose to exert effort but failed to reach the chosen effort level, or if the participant did not make a choice on time, the trial was deemed failed, and they received 0 credits. This was implemented to keep the paradigm incentive-compatible and to preserve the intended trade-off between a small, guaranteed payoff and a larger, contingent payoff. Without this rule, participants could have selected the costly option without actually following through, and choices would no longer reflect a genuine willingness to incur costs.

Pro-environmental time task. The time task follows the design of the effort task, with the only difference being the cost (instead of effort, it is a waiting period), and was inspired by research on intertemporal choice[40,41]. On each trial, participants made a choice between no waiting time for 1 credit and a varying (individually calibrated 5 levels, e.g. 3, 6, 9, 12, 15 sec [possible range: 3-27 secs]) time investment for a higher reward (varying; 2, 4, 6, 8, 10 credits; see Fig. 1B and SI Fig. S2B for a decision tree). When they chose the waiting time option, they had to endure this waiting period, which was indicated on the screen using the same bar used to indicate the effort at the effort task, which was filling proportionally to the time passing. During the waiting time, participants could not engage in other activities. Importantly, the trial length was dependent on the choices the participant made, which makes the time investment real and not hypothetical – participants could finish the experiment faster if they decided to take the less costly option. If participants selected the waiting time, they could press a button to skip it (for consistency with the effort task, where they have the option of not reaching the required level of effort). This would result in a failed trial, and they would get no reward for it either (as in the effort task, when failing to attain the threshold).

There were six counterbalanced blocks (three effort and three time blocks), each containing 50 trials, which participants completed with breaks in between. Before the start of the blocks, participants experienced five practice trials for each task to ensure they understood the instructions. During the main blocks, the participant and the experimenter were sitting in separate rooms, but with an open door, so the experimenter could still see if the participant was engaging in the task, but without seeing the content on their screen and, respectively, their choices. The session ended with repeating the procedure, where the participants experienced each level of time and effort and answered follow-up questions in a short debriefing.

**Statistical analysis**
Our analysis plan was preregistered on the Open Science Framework (https://osf.io/bksf9; on 16.03.2022). Deviations are explained below, and analyses that were not preregistered are explicitly reported as exploratory. Data were analysed in R[42].

**Choice, decision time, and force**
To analyse participants' choices, we used logistic mixed-effects models, and to analyze the decision time and force data, we used linear mixed-effects models (see SI, Section 2.6, for packages used and predictor scaling). All models included the following variables: recipient (i.e., self or environment), cost level (effort or time, respectively), reward level, and all their interactions, and additionally included trial number as a covariate. Furthermore, to allow for maximally defined error structures[43], we added a random intercept and random slopes for all independent variables and their interactions, nested within participants as a random factor (see SI, Section 3.4.1 for all model formulae). Due to convergence issues, we followed the suggestions outlined in previous work[44] by first centring predictors, increasing the maximum

number of iterations, and, lastly, by successively removing the random error term that explained the least variance until convergence was achieved. In addition, for the analyses of choice and response time, we excluded trials where participants failed to decide within the 3-sec window (exclusion rates: effort task: self = 0.16%, environment = 0.36%; time task: self = 0.18%, environment = 0.42%). Furthermore, we removed trials where participants did not select the effortful option for analyses of the force data.

**Deviations from the preregistration.** We had not preregistered the inclusion of trial number as a covariate. We decided to do so to control for fatigue, as participants rated the effort and time levels post-experiment as more demanding compared to prior (see Tables S14 and S15). Additionally, we preregistered to include only participant code in the random term but opted for using a more complex random structure to follow best practices[43]. The preregistered analyses without trial number and with the simpler random term are suboptimal as they inflate the hypothesized relationship, but are nevertheless reported in SI, Section 3.7.1 for transparency.

**Climate change beliefs**
To assess the effects of climate change key beliefs on the difference in discounting, we calculated the difference in discounting parameters for environmental outcomes vs self-outcomes (k self–k env; see computational modelling section of the main manuscript for definition of parameters). We opted to use robust multiple regression from the MASS package[45] due to the presence of outliers and violations of normality of the residuals. Robust regressions are less sensitive to violations of assumptions such as normality and homoscedasticity, thus ensuring that extreme values do not unduly influence our findings.

**Deviations from the preregistration.** Conducting our preregistered analysis, which included creating an average score of the five subscales, was suboptimal due to the psychometric properties of the climate change key beliefs and the highly skewed distribution of some items (see SI, Section 1 for detailed explanation). For transparency, we report the results of the originally preregistered analysis in SI, Section 3.7.2. To nevertheless investigate the association between climate change key beliefs and effort and time discounting, we opted for robust multiple regression with each of the different beliefs as a predictor.

**Support for climate policies**
To investigate the relationship between the difference in discounting of self and environmental rewards and support for climate policies, we used Kendall's rank correlation coefficient.

**Deviations from the preregistration.** We preregistered using a linear regression to test this association but opted for a non-parametric test due to the non-normal distribution of the data and the presence of outliers. The results of the preregistered linear regression are reported in SI, Section 3.7.3.

**Other interpersonal differences**
We conducted exploratory analysis (i.e., not preregistered) using Kendall's rank correlation coefficient to investigate the associations between several interpersonal differences and the difference in discounting tendencies.

**Confirmation of null results with Bayesian statistics**
For all frequentist analyses yielding null effects, we conducted Bayesian analyses to complement the frequentist results. The Bayesian models mirrored the corresponding frequentist models (e.g., Bayesian linear mixed models for null effects observed in frequentist LMMs), ensuring that the inference was based on the same statistical structure. We report Bayes Factors (BF$_{10}$) alongside the null findings. Detailed information on the model specification and packages used can be seen in SI, Section 2.7.)

**Computational modelling**
We employed a hierarchical Bayesian modelling approach that simultaneously estimated decision processes at a group- and individual-level to more accurately capture behavioural variation[46] caused by discounting of rewards by effort vs time. To investigate the subjective influence of these two types of cost on decision-making for each individual, we ran a set of 12 computational models with hierarchical Bayesian estimation for each task. Each of these models is based on the assumption that the probability of choosing an option is a function of its subjective value (SV), which depends on the reward R and the cost C associated with the option. Based on previous work on discounting of effort and time[12,16,19,27,47], we tested three different discounting functions:

$$\text{Linear}: SV = R - k * C$$

$$\text{Parabolic}: SV = R - k * C^2$$

$$\text{Hyperbolic}: SV = R * \frac{1}{1 + k * C}$$

In each of these three functions, the discounting parameter k governs the degree to which rewards are discounted by costs (either effort or time), with higher values of k indicating higher discounting (i.e., higher devaluation of the reward due to the cost).

To transform subjective values into choice probabilities (P), we used the softmax function:

$$P(\text{costly option}) = \frac{1}{1 + \exp(-\beta(SV \text{ costly option} - SV \text{ no cost option}))}$$

Here, the parameter β governs the stochasticity of choices: If β is high, the option with the higher SV is chosen more consistently. If β is close to zero, choices are more random (i.e., the option with the lower SV also has a substantial probability of being chosen). Description of the parameterization and priors used can be found in SI, Section 2.8.1.

We compared models with a separate (self and environment) or a joint cost discounting parameter k and separate or joint stochasticity parameter β. Each of these four combinations was separately tested for the linear, hyperbolic, and parabolic discounting function, resulting in 12 distinct models (four parameter combinations multiplied by three discounting models).

**Model estimation and selection.** For model estimation, we used the rstan package in R[48] followed by best practices with the hBayesDM package[49]. We ran four independent Markov Chain Monte Carlo (MCMC) chains with 2000 iterations for each candidate model, discarding the first 1000 iterations per chain for fine-tuning. The remaining 1,000 iterations per chain were retained, yielding a total of 4000 valid posterior MCMC samples. All the models reported in the main manuscript converged with $\hat{R} < 1.05$. For model selection, we compared the Leave-One-Out Information Criterion[50] (LOOIC), which examines model performance while penalizing model complexity, thus preventing overfitting. Lower LOOIC scores indicate better prediction accuracy.

**Model validation and parameter recovery.** We performed posterior predictive checks and a parameter recovery analysis to validate the models, using data generated from the best-fitting model and simulated participants. This synthetic dataset was created by randomly drawing individual parameter values from the posterior group-level ground truth estimates derived from the winning model. First, we simulated choice probabilities to assess whether the generated data replicated key patterns observed in participant behaviour, specifically how choices varied based on cost (i.e., effort and time) and reward (see SI, Section 2.8.2). Next, we examined group-level parameter recovery by determining whether the

95% highest density interval (HDI) of the true group-level parameter distribution encompassed the mean estimates of the simulated group-level parameters (see SI, Section 2.8.3). Finally, we assessed subject-level parameter recovery by computing the correlation between the true and simulated subject-level parameter estimates (see SI, Section 2.8.3).

### Reporting summary
Further information on research design is available in the Nature Portfolio Reporting Summary linked to this article.

## Results
### Higher motivation to choose the costly option for self- vs environment trials
To analyse the choices in the effort task, we used a logistic mixed-effects model with choice as the outcome variable. We found that participants were less likely to select the effortful option for the environmental trials than for trials where rewards were paid out to themselves (OR = 0.32, 95% CI [0.15, 0.70], $p = 0.004$; see Fig. 1A). Additionally, participants were less likely to select the effortful option with increasing levels of effort (OR = 0.15, 95% CI [0.12, 0.18], $p < 0.001$) and more likely to select it with increasing levels of reward (OR = 4.68, 95% CI [3.70, 5.93], $p < 0.001$). There was a significant recipient x reward interaction (OR = 0.75, 95% CI [0.60, 0.93], $p = 0.009$), showing that the effect of reward was smaller for environmental trials, or in other words, the difference between self and environmental decisions was larger at the higher levels of reward. There was also an effort x reward interaction (OR = 0.83, 95% CI [0.77, 0.90], $p < 0.001$), indicating that with increasing effort levels, rewards were less incentivizing. Figure 1A and SI Fig. S8A illustrate these effects, and the whole model output is documented in SI Table S2.

To analyse the choices in the time task, we used the same approach as in the effort task, i.e., a logistic mixed-effects model with choice as the outcome variable. This revealed that participants were less likely to select the waiting option with increasing time demand (OR = 0.30, 95% CI [0.23, 0.39], $p < 0.001$), and more likely to select it with increasing reward offered (OR = 7.05, 95% CI [4.54, 10.93], $p < 0.001$). There was no statistically significant effect indicating individuals were more likely to select the waiting option for themselves overall (OR = 0.44, 95% CI [0.18, 1.07], $p = 0.069$), but there was a significant recipient x reward interaction (OR = 0.79, 95% CI [0.65, 0.96], $p = 0.020$), indicating such difference was seen when the rewards offered were higher. Figure 1B and SI Fig. S8B illustrate these effects, and the whole model output is documented in SI Table S3.

Notably, 40.54% of the participants showed very little variability in their task behaviour for the time task, selecting the waiting time option on >95% of the trials (compared to 6.76% in the effort task), which may indicate the presence of a ceiling effect. As per our preregistration, we also reran the models for both tasks, excluding such participants. The results were largely unchanged, with the only notable difference being that the effect of the recipient became significant in the time task, indicating lower willingness to select the waiting option when the rewards offered were for the environment (OR = 0.29, 95% CI [0.12, 0.72], $p < 0.001$; see SI Section 3.4.2. for the full output of the models). To explore potential social desirability effects, we also ran the models excluding participants who wrote anything other than "no" in the open-ended question asking whether they felt observed when making their choices (excluded $n = 11$). The results remained largely the same, with the only notable difference being that the effect of the recipient became significant in the time task (see SI Section 3.4.3 for the full output of the models).

### Participants were equally fast in making decisions for self- and environmental outcomes
We used linear mixed models using the duration it took participants to make a decision as the dependent variable. For the effort task, there was no statistically significant difference in decision duration for self vs environmental trials ($b = 0.01$, 95% CI [-0.01, 0.03], $p = 0.466$, $d = 0.02$), confirmed by Bayesian analysis (BF$_{10}$ = 0.171), indicating moderate evidence for the

null hypothesis. There was an effect of effort such that participants were taking longer to make decisions on trials involving higher effort ($b = 0.06$, 95% CI [0.05, 0.07], $p < 0.001$, $d = 0.18$). We also found an effort x reward interaction ($b = 0.02$, 95% CI [0.01, 0.03], $p < 0.001$, $d = 0.06$), indicating that with increasing reward levels, the effect of effort was larger (i.e., participants took longer to decide; see Fig. 2C and SI Table S4 for the full model output).

Similarly, for the time task, there was no statistically significant difference in decision duration for self vs environmental decisions ($b = 0.01$, 95% CI [-0.01, 0.03], $p = 0.341$, $d = 0.03$), confirmed by Bayesian analysis (BF$_{10}$ = 0.180), indicating moderate evidence for the null hypothesis. There was an effect of time, such that participants were taking longer to make a decision on trials involving longer waiting times ($b = 0.04$, 95% CI [0.03, 0.05], $p < 0.001$, $d = 0.12$), an effect of reward, showing that participants were faster on trials offering a larger reward ($b = -0.02$, 95% CI [-0.03, -0.01], $p < 0.001$, $d = 0.06$), and a time x reward interaction ($b = 0.01$, 95% CI [0.00, 0.01], $p = 0.012$, $d = 0.02$), such that with increasing reward levels, the effect of time was larger, i.e., participants were taking longer in making decisions for offers with higher rewards (see Fig. 2D and SI Table S5).

### No statistically significant difference in overall force expended, but longer duration above the force threshold for self-outcomes
Next, we investigated the overall force expenditure (calculated as the area under the curve; AUC) for self vs environment decisions using a linear mixed model with AUC as the dependent variable. There was no significant difference between self-benefitting and environmental decisions ($b = -0.003$, 95% CI [-0.007, 0.001], $p = 0.107$, $d = 0.06$), and Bayesian analysis (BF$_{10}$ = 0.401) was consistent with this finding, suggesting weak support for the null hypothesis. There was an effect of effort level ($b = 0.070$, 95% CI [0.067, 0.073], $p < 0.001$, $d = 1.31$) and reward ($b = 0.004$, 95% CI [0.003, 0.006], $p < 0.001$, $d = 0.08$) such that the higher the effort level, and the higher the reward, the higher the AUC; see SI Table S6 for the whole model output. Moreover, when participants chose to put in the effort, there were no significant differences between the recipients in how often they successfully squeezed to the required effort level and received the reward (mean success rate self = 94.7%, environment: 93.8%, paired t-test, mean individual differences = 0.87%, 95% CI [-0.51%, 2.24%], $t(72) = 1.255$, $p = 0.213$).

However, exploratory analysis using linear mixed models looking at the amount of time pressed with the required degree of force (i.e., above the threshold line indicating success) as the dependent variable showed that people were pressing shorter with the required degree of force on trials for the environment ($b = -0.043$, 95% CI [-0.074, -0.011], $p = 0.008$, $d = 0.10$), see SI Table S7 for the whole model output.

### Computational modelling reveals parabolic effort discounting and hyperbolic temporal discounting with separate parameters for self and environmental outcomes
For the effort task, the parabolic model with two discounting parameters and two stochasticity parameters (2k2ß) was the best fitting model (lowest LOOIC = 6702; Fig. 3A), with all parameters showing satisfactory parameter recovery (Fig. 3B and SI Fig. S4B). This suggested that participants' valuation processes were distinctive when computing values for self-benefitting vs pro-environmental actions and indicated they discounted the outcomes parabolically by physical effort (Fig. 3C). Additionally, the discounting was significantly stronger for environmental rewards compared to rewards for oneself (Wilcoxon signed-rank test: $V = 1009$, $p = 0.021$, one-tailed), with substantial interindividual variability, as indicated by Fig. 3D and the standard deviation of the difference in discounting tendencies for self vs environmental outcomes which was around 5 times larger than the standard deviation of the discounting of self rewards (SD of k diff = 0.31; SD of k self = 0.06). This indicates that while the average effect indicated stronger discounting for environmental rewards, this average conceals considerable heterogeneity: some participants showed the expected pattern of steeper environmental discounting, whereas others showed little difference or even found environmental rewards more motivating.

For the time task, the 2k2ß hyperbolic model had the best model performance (lowest LOOIC = 4262, Fig. 3D), with all parameters showing satisfactory parameter recovery (Fig. 3E and SI Fig. S4D). This suggested that participants had distinctive valuation processes when computing values for self-benefiting and pro-environmental decision options, and discounted rewards hyperbolically by waiting time (Fig. 3F). Although the mean difference indicated that the discounting was not significantly stronger for environmental rewards (Wilcoxon signed-rank test: $V = 1527$, $p = 0.775$, one-tailed, confirmed by Bayesian analysis showing moderate evidence for the null hypothesis, $BF_{10} = 0.126$), the standard deviation of the difference in discounting tendencies for self vs environmental outcomes was substantial (2.33 times larger than the standard deviation of the discounting of self rewards; SD of k diff = 0.53; SD of k self = 0.23), with some participants showing higher discounting for themselves and others showing higher discounting for the environment (see also Fig. 3H).

Next, we investigated how the parameters in the effort task were associated with those in the time task, using Kendall's rank correlation coefficient. We found that self discounting parameters in both tasks (k effort self and k time self), as well as the environment discounting parameters (k effort env and k time env), were significantly correlated across the types of costs (k effort self and k time self: $\tau = 0.25$, $p < 0.001$, k effort env and k time env ($\tau = 0.33$, $p < 0.001$). Importantly, the differences between the discounting parameters for self and the environment (k effort diff and k time diff) were also significantly correlated ($\tau = 0.57$, $p < 0.001$). This indicates that participants who were more willing to incur effort costs for pro-environmental outcomes (relative to self-outcomes) were also more willing to incur time costs for pro-environmental outcomes (relative to self-outcomes).

Lastly, exploratory modelling analyses showed that the models incorporating additional reward sensitivity parameters next to effort discounting and choice stochasticity parameters provided a better fit (see SI, Section 3.8). However, they showed poor identifiability, making them unsuitable for subsequent analysis. To address this, we explored further models containing reward sensitivity and effort discounting parameters but without the choice stochasticity (ß) parameter. These models showed good fit and identifiability, but due to the exploratory nature of this analysis, we documented them only in the SI and focused on the preregistered winning model for the subsequent analyses.

### Climate change key beliefs do not predict the difference in discounting for self vs environmental outcomes

To test the hypothesis that climate change key beliefs are associated with the difference in discounting, we used a robust multiple regression including the five climate change key beliefs (i.e., climate change belief certainty, belief about anthropogenic cause, collective efficacy, harm extent, and harm timing) as predictors and the differences in effort and temporal discounting for self and environmental outcomes in both tasks as outcomes. None of the key climate change beliefs significantly predicted differences in discounting in either task, and Bayesian analyses supported these findings. For the effort task, no predictors were statistically significant: harm timing ($p = 0.196$,

$BF_{10} = 0.042$, strong evidence for the null), harm extent ($p = 0.505$, $BF_{10} = 0.025$, strong evidence for the null), belief certainty ($p = 0.052$, $BF_{10} = 0.849$, anecdotal evidence for the null), anthropogenic cause ($p = 0.516$, $BF_{10} = 0.087$, moderate evidence for the null), and collective efficacy ($p = 0.917$, $BF_{10} = 0.023$, strong evidence for the null). Similarly, for the time task none of the predictors showed statistically significant effects: harm timing ($p = 0.499$, $BF_{10} = 0.064$, strong evidence for the null), harm extent ($p = 0.177$, $BF_{10} = 0.066$, strong evidence for the null), belief certainty ($p = 0.095$, $BF_{10} = 0.085$, moderate evidence for the null), anthropogenic cause ($p = 0.758$, $BF_{10} = 0.056$, strong evidence for the null), and collective efficacy ($p = 0.628$, $BF_{10} = 0.051$, strong evidence for the null). The full model output can be seen in SI Table S12 for the effort task model and Table S13 for the time task model.

### Support for costly climate policies is associated with the discounting difference between self and environmental outcomes

Next, we tested our hypothesis that higher discounting of effort and time for environmental outcomes compared to self-outcomes would be associated with lower support for costly climate change mitigation policies. The analysis revealed a significant correlation between the difference in discounting (i.e., k self–k env) in both tasks and policy support (Kendall's rank correlation coefficient: effort: $\tau = 0.26$, $p < 0.001$, time: $\tau = 0.13$, $p = 0.048$). The higher the difference in discounting (i.e., the less participants chose to invest effort or time for environmental compared to self-benefitting outcomes), the less participants were willing to support costly climate policies.

### Exploratory analysis of interpersonal differences and their relationship to discounting for self and environmental outcomes

Observing substantial inter-individual differences in discounting prompted us to explore the relationship between interpersonal differences in self-report measures and the differences in discounting tendencies (see Fig. 4). As these interpersonal correlations were examined in an exploratory manner, results are reported uncorrected for multiple comparisons and should be interpreted with appropriate caution. Our analysis showed that the only variable correlating with the discounting difference in both tasks was political orientation (Kendall's rank correlation coefficient: effort task: $\tau = 0.33$, $p < 0.001$, time task $\tau = 0.22$, $p = 0.012$), such that more liberal people showed lower discounting of environmental rewards (i.e., higher environmental motivation). For the effort task, we additionally observed a positive relationship with self-reported pro-environmental behaviour. Participants who reported more pro-environmental behaviour showed higher environmental motivation in the experiment ($\tau = 0.25$, $p < 0.002$). We also saw a positive relationship with biospheric values ($\tau = 0.30$, $p < 0.001$) and age ($\tau = 0.23$, $p = 0.004$), and a negative relationship with egoistic values (i.e., higher egoistic values were associated with less pro-environmental motivation in the task; $\tau = -0.16$, $p = 0.049$). For the time task, there was an association with education ($\tau = 0.20$, $p = 0.031$), such that more educated participants were discounting environmental rewards less. Importantly, participants' subjective financial situation was not associated with the differences in discounting in either of the tasks.

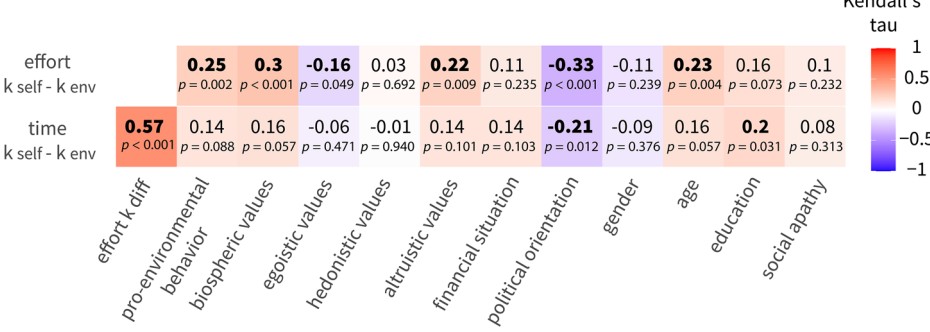

**Fig. 4 | Difference in discounting of environmental rewards vs self rewards for the time and effort task and interpersonal differences.** High values for the difference indicate higher pro-environmental motivation. Bold numbers indicate significant relationships tested by Kendall's rank correlation coefficient, two-sided, and not corrected for multiple comparisons ($n = 74$ participants).

## Discussion

In this preregistered experiment, we investigated how individuals discount time and effort when making self-benefiting vs pro-environmental decisions. Building on extensive research on motivation and decision-making, we adopted a paradigm previously used in the domain of prosocial decision-making[16,17,19] to investigate effortful decision-making benefiting the environment relative to the self. We additionally developed a second paradigm that closely followed the design of the effort task but used time instead of effort costs. Our analyses show that individuals were generally more reluctant to incur costs for environmental benefits than for self-benefiting outcomes. However, there was substantial individual variability in this tendency. Participants who were less willing to incur effort and time costs for the environment were also less likely to support costly climate policies, and exploratory analysis revealed they were more likely to be politically right-leaning. By directly comparing pro-environmental with self-benefiting actions within the same paradigm, our study highlights critical motivational asymmetries and opens promising directions for future research.

First, we show that motivation increases as potential rewards increase and behavioural costs decrease. This finding not only aligns well with previous research on environmental decision-making[4,51], but is also consistent with the fundamental principles of reward valuation, where greater incentives can partially offset the perceived aversiveness of effort and time costs. Importantly, when higher rewards were at stake, the gap between self-benefitting and environmental motivation widened, highlighting that increases in the potential environmental benefits are less incentivizing than equivalent increases in personal monetary rewards. This finding further suggests that getting people to accept the behavioural costs associated with climate action might require more than simply emphasizing the large environmental benefits. Instead, they highlight the central role of perceived costs as barriers to climate action. Prior research has shown that as costs increase, environmental attitudes become less predictive of people's behaviour[34,52,53]. This underscores the limitation of relying solely on strengthening climate change beliefs or environmental attitudes to drive behaviour change, without also addressing the behavioural costs that often hinder action. Reducing these costs – through measures such as default nudges, convenient infrastructure, or policies that integrate sustainable options into daily routines – may be essential for enabling more consistent pro-environmental behaviour.

Interestingly, while participants were overall less likely to select the effortful option for environmental rewards, when they did commit to an effortful environmental choice, they energized their actions to the same extent as they did for self-benefiting choices (the overall force expended over the three-second window did not differ significantly; note that the Bayesian analysis favoured the null hypothesis but only weakly). Overall, we did not find evidence that participants exhibit superficial pro-environmental motivation, such as opting for an effortful action but executing it with less vigour, which has been shown in research on prosocial motivation[16,17,19]. However, an exploratory analysis also showed that participants sustained their force above the required threshold for a longer duration for self-trials than environmental trials. This indicates that while the overall amount of force was did not differ significantly, the distribution of the force may have been more strategic in self trials. Although this nuance in force distribution did not alter immediate outcomes (both self and environmental choices maintained similar success rates), it may point to a subtler form of self-serving bias that might become more consequential in more demanding or sustained real-world contexts.

Our computational modelling analyses showed distinct discounting patterns for effort and time. Effort discounting followed a parabolic function, consistent with prior research on effort-based decision-making[14,16,17,19,27], whereas temporal discounting was best captured by a hyperbolic function, a well-established characteristic of temporal decision-making[12]. A parabolic form implies that the subjective devaluation of rewards grows disproportionately as effort demands increase – the subjective value drops slowly when the required level of effort is low but accelerates when the effort required gets higher. By contrast, a hyperbolic form implies that the subjective value drops sharply even for short waiting times, then flattens as delays lengthen, consistent with the immediacy effect and diminishing marginal opportunity costs of additional delay. Although experiential delays in the range of seconds, as used in our paradigm, differ from the longer horizons more commonly studied in the temporal discounting research, both literatures converge on hyperbolic discounting as the characteristic functional form, suggesting continuity across timescales[12,14,41,54]. Taken together, these functional differences suggest that effort and time costs are processed through distinct cognitive and physiological mechanisms. This aligns well with evidence from neuroimaging findings reporting the engagement of distinct brain areas for each type of cost[9], and highlights the need to study these costs separately in future decision-making research.

Furthermore, our winning models incorporated two separate discounting factors (i.e., one for self and one for environmental outcomes), indicating that participants varied in their discounting tendencies. On average, participants showed steeper discounting for themselves in the effort task, while the difference in the time task was not statistically significant (which may be partly due to a ceiling effect in the time task). Nevertheless, we observed substantial interindividual differences, with some participants showing higher motivation for the environment while others showed higher motivation for themselves, highlighting the utility of this modelling approach in capturing motivational variation.

While our experimental paradigm focuses specifically on the cost-incurring dimension of pro-environmental action, we acknowledge that this represents only one facet of the broader motivational landscape. It is important to recognize that climate action is influenced by factors operating at multiple levels. At the individual level, cost-benefit evaluations play a role, but so do psychological and social drivers such as values, identity, or perceived norms[26,55]. At the structural level, the availability of supportive infrastructure can facilitate or hinder pro-environmental choices, while at the national level, the broader context, such as the level of development of the country people live in or its fossil fuel reliance, can also influence people's willingness to act[26,56–58]. Under certain conditions, these factors could also reduce or even eliminate the perception of costs, such that pro-environmental behaviours may be experienced as intrinsically desirable rather than as costly sacrifices. Importantly, our findings show that participants invested considerable effort and time on behalf of the environment in absolute terms: on average, they chose the costly option in 60.4% of trials in the effort task and 78.9% of trials in the time task. This demonstrates that, despite the relative inferiority of environmental rewards, participants were willing to incur personal costs for pro-environmental outcomes. Future research could, thus, build on these findings by using our paradigm to examine how individual, social, and structural factors interact with cost-incurring decisions, how subjective perceptions of costs (e.g., boredom) influence willingness to act, and how interventions might help reframe pro-environmental actions from perceived sacrifices into intrinsically valued behaviours

Contrary to our preregistered expectations, climate change beliefs were no significant predictors of the difference in discounting parameters. Bayesian analyses largely supported these null effects, providing moderate to strong evidence for the null hypothesis for both tasks, with the exception of climate change belief certainty in the effort task, where the evidence for the null hypothesis was less strong but still favouring it. There are several potential explanations for the overall lack of associations. First, there is a growing body of research showing that attitudes do not necessarily translate to action, especially when costs are high[52,53]. Beliefs can be endorsed at little or no personal cost. For instance, agreeing with the view that climate change is a serious problem can be largely symbolic and will not require individuals to act upon this view or to sacrifice resources. As a result, these beliefs may capture general attitudes rather than the motivational processes that guide cost-incurring decisions. Second, it is important to note that our sample reported overall high climate change beliefs, with 77% of participants selecting the highest degree of certainty that climate change is happening (see SI Fig. S3). The related lack of interindividual variation may thus be an

alternative explanation of why we did not find a significant relationship between beliefs and behaviour. That said, other research has shown little difference between sceptics and believers in terms of engaging in costly actions[59]. Relatedly, our study was primarily powered to detect differences in discounting between self and environment conditions; thus, if the true association between climate change beliefs and costly behaviour is very small, we may have been underpowered to detect it reliably. This notwithstanding, our study supports prior findings suggesting that climate change beliefs, while important for signalling general attitudes, do not necessarily translate into actual willingness to incur personal costs. This underscores the value of focusing on predictors more tightly linked to costly pro-environmental behaviour.

By contrast, we confirmed our preregistered hypothesis that participants who exhibited higher discounting of environmental rewards compared to self-benefitting rewards were also less likely to support costly climate policies, as measured with a scale we developed to focus on trade-offs in policies. One reason may be that policy support inherently entails an acceptance of potential personal or societal costs, making it more closely aligned with cost-sensitive choices in our paradigm. Relatedly, by linking laboratory-based measures of motivation with policy attitudes, our findings suggest that individuals' willingness to endure costs for pro-environmental actions in controlled settings could relate to their support for system-level actions like climate mitigation policies, even if they came with costs. Given that effective climate change mitigation depends on both individual choices and collective policy measures, this finding highlights parallels between the personal cost aversion at the individual choice level and support for measures at the societal level – an insight that may help explain why ambitious climate policies often struggle to gain broad support.

In an exploratory analysis, we also examined how several socio-demographic, psychological, and behavioural factors are related to differences in time and effort discounting. This revealed that individuals with a more liberal political orientation exhibited lower discounting rates for environmental outcomes relative to self-outcomes (i.e., higher pro-environmental motivation). We further observed a relationship between the difference in effort discounting and people's values, such that participants who reported higher biospheric and altruistic values, and lower egoistic values, showed lower discounting of pro-environmental outcomes in the effort task. These exploratory findings reinforce the role of underlying value and ideological systems in the willingness to incur personal costs to benefit the environment and are supported by a large body of literature on pro-environmental actions[4,51,60–62].

Beyond the conceptual findings, our study is also promising in terms of some methodological insights, as both tasks showed utility in investigating the role of costs in influencing pro-environmental motivation. The observed high correlation between the difference in discounting for self vs environmental outcomes in the effort task and time task, suggests that both tasks tap into a common underlying construct: the degree to which individuals prioritize environmental vs self-benefitting outcomes when costs are involved. Moving beyond hypothetical scenarios, these tasks employ a multi-trial structure with real costs and benefits for both self and environmental outcomes, enabling the systematic manipulation of several decision-making factors. This allows researchers to conduct a precise investigation of their influence on pro-environmental actions, making it a promising tool for future research. This aspect may be particularly relevant for adaptations in neuroscientific and eye-tracking studies, as the approach enables more precise investigations into the neuro-cognitive mechanisms underlying environmental decision-making[63,64]. Beyond these applications, our paradigm could also provide a means to experimentally testing interventions aimed at increasing people's willingness to accept non-financial costs such as effort or time for environmental benefit. These types of cost are under-represented in current intervention research, despite being highly relevant for everyday sustainable behaviours. Using the paradigm to evaluate such strategies in a controlled setting could therefore help identify promising approaches before implementing large-scale interventions in the real world.

It will, however, first be of central importance to further validate how the laboratory-experimental setup emulates discounting of environmental benefits under everyday life conditions, and how our findings can be generalized to decision-making outside of the lab. While the correlation with self-reported pro-environmental behaviour was rather low (and significant only for the effort task), the consistency of the associations with policy support, personal values, and political orientation, which are established correlates of climate action[4,51,60–62], suggests our paradigm picks up meaningful interindividual variation. The overall consistency of findings across both tasks highlights their potential for studying the impact of effort and time costs on pro-environmental decision-making and provides a foundation for further methodological improvements; we also hope the tasks and their respective utility provided inspiration for their targeted use in follow-up research. To encourage researchers to use it, we have published all the scripts required for running the tasks and analyzing the data (https://osf.io/aj38k/files).

## Limitations

A potential inherent limitation of our design is that the tasks were completed in the laboratory, which may increase social desirability effects. We attempted to minimize these by placing participants in a separate room from the experiments for the main part of the experiment, and excluding people who wrote anything other than "no" to questions about feeling observed did not change the results meaningfully. However, prior work on pro-environmental motivation has documented differences between lab and online behaviour, such that the self-environment differences were larger in the online setting[28], so a potential future direction could be to adapt these paradigms for online use, or manipulate social desirability, and assess whether the magnitude of the effects will differ. Another important point is that our policy support measure does not allow us to determine whether the costs described in the items were perceived as personally relevant or primarily of societal relevance. Future research is needed to explicitly assess the personal vs societal relevance of policy costs and examine how this distinction shapes people's support. It is also important to mention that the time task exhibited a ceiling effect, with some participants showing no variability in their behaviour (i.e., selecting the waiting option on every trial). Although participants were told that the duration of the experiment depended on their choices, it remains possible that by committing to a laboratory session, they already anticipated spending a considerable amount of time in the lab, which may have reduced the subjective salience of time costs. This lack of variability limits the sensitivity of the time task in capturing individual differences in discounting tendencies. A next step would be to refine the time task, for example, by increasing the waiting durations, as it holds potential for broader application in the field of motivation. Relatedly, our time–effort calibration was designed to align the lowest and highest effort levels with corresponding waiting times, interpolating intermediate values. Given that effort and time follow different nonlinear discounting shapes, this approach is necessarily an approximation. Future work could increase precision by tracing multiple indifference points per participant (adaptive staircases) to estimate full indifference curves or by using conjoint/multi-attribute models that weight time and effort jointly. Lastly, although we carefully emphasized the strong track record of the selected organization, we cannot rule out the possibility that participants' willingness to choose the environmental option was influenced by their subjective perceptions of the organization, like e.g., the degree of trust they placed in it or the efficacy of measures suggested.

## Conclusion

This study offers insights into the processes underlying pro-environmental motivation and presents a promising approach for measuring pro-environmental motivation under controlled experimental laboratory conditions. By integrating real effort and waiting time costs with concrete environmental consequences—where participants' choices directly impact $CO_2$ reduction—our paradigm captures real pro-environmental decision-making and adds to a small but growing literature using incentivised

paradigms to investigate pro-environmental motivation[4,25,27]. While our findings show that behavioural costs act as barriers to higher motivation, the substantial inter-individual variation captured by our paradigm, and the fact that some people do engage in costly climate actions, indicate that these barriers are not insurmountable. Pinpointing the factors that lead individuals to accept time and effort costs could be crucial for designing interventions and policies that support sustained engagement in pro-environmental behaviour.

## Data availability
The data are publicly available on OSF: https://osf.io/aj38k/files.

## Code availability
The code for running the experimental paradigm, for analysing the data, and reproducing the figures is available at OSF: https://osf.io/aj38k/files.

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

## Acknowledgements
The authors thank Tamás Auer, Marie-Therese Mang, Fabian Ottolin, and Timon Schnauffer for their help with data collection and Claudia Massacessi for helpful discussions about the computational modelling analyses. We would also like to thank our participants for their efforts in the task, as a result of which we donated 514,93 Euro to "Verein Regenwald der Österreicher" [Austrian Rain Forest Society] to be used towards their $CO_2$ offsetting and biodiversity scheme. This research was partly funded by the Austrian Science Fund (FWF): "DK Cognition and Communication 2": W1262-B29 [10.55776/W1262], "Neuronal circuits in health and disease": COE16 [10.55776/COE16], and "The role of incentives and uncertainty in prosocial behaviour", to CL [10.55776/PAT1936023]. LZ was partially supported by the FWF [10.55776/M3166] and Wellcome Trust [228268/Z/23/Z]. The funders had no role in study design, data collection and analysis, decision to publish or preparation of the manuscript.

## Author contributions
B.T.—conceptualisation, methodology, software, validation, formal analysis, investigation, resources, data curation, visualization, writing—original draft, project administration; L.Z.—conceptualisation, methodology, formal analysis, resources, writing—review & editing, supervision; L.L.—methodology, formal analysis, writing—review & editing; J.N.—formal analysis, writing—review & editing; K.C.D.—conceptualisation, supervision, writing—review & editing; S.P.—conceptualisation, writing—review & editing; P.F.—conceptualisation, writing—review & editing; C.L.—conceptualisation, methodology, resources, supervision, funding acquisition, writing—review & editing.

## Competing interests
The authors declare no competing interests.
