## [Transparent Peer Review file · Communications Psychology]

Effort and time costs influence motivational asymmetries in self-benefitting vs pro-environmental decisions

Corresponding Author: Professor Claus Lamm

Version 0:

Decision Letter:

Dear Professor Lamm,

Thank you for submitting your manuscript titled "Effort and time costs influence motivational asymmetries in self-benefitting vs pro-environmental decisions" to Communications Psychology. We have given the paper our careful consideration and find it of potential interest. However, due to certain shortcomings we are concerned that sending the current manuscript out to review could lead to unnecessary delays and quite possibly an undesirable outcome of the review process.

It is our policy that authors must disclose all deviations from the preregistered protocol and explain the rationale for deviation (e.g., flaw, feasibility, suboptimality). In cases of deviation from the preregistered analysis plan for reasons other than fundamental flaw or feasibility, the originally planned analyses must also be reported. You can find our full policy on preregistration here: <https://www.nature.com/commpsychol/submit/preregistration>

For hypothesis 3, you note a switch to Kendall's rank coefficient instead of linear regression due to the non-normal distribution of the data. From this description, it's unclear whether the planned analysis is fundamentally flawed or infeasible. Please provide more details on the rationale behind the deviation and include the planned analysis unless fundamentally flawed or infeasible.

The deviations (and rationale) should be explicitly mentioned in the main article file rather than the supplementary material. Also please specify in the main article file whether the analyses reported in the section "Interpersonal differences and their relationship to discounting differences for self and environmental outcomes" are preregistered or exploratory.

We would therefore like to invite you to revise your manuscript to address these concerns before we make a final determination on whether to send your manuscript for external review.

We shall hope to receive your revised version as soon as you are able to complete the suggested revisions. If something similar is published in the interim we will have to consider the impact it has on the novelty of a revised manuscript.

If you anticipate a delay of more than four weeks, please let us know. Should your manuscript be substantially delayed without notifying us in advance and your article is eventually published, the received date may be that of the revised, not the original, version.

We also ask that you ensure your manuscript complies with our editorial policies and reporting requirements.

To that end, we require revised manuscripts to be accompanied by a completed item: a reporting summary that collects information on study design and procedure.

- <https://www.nature.com/documents/nr-reporting-summary.pdf>>Nature Research Reporting Summary

Your revised manuscript can only be sent to referees if the checklist is completed and uploaded with the revision.

If you are not interested in submitting a suitably revised manuscript in the future please let me know immediately so we can close your file. If you have any questions, please contact me.

Please use the link below when you are prepared to resubmit.
Link Redacted

Thank you for your interest in Communications Psychology.

Best regards,
Mael Lebreton

Mael Lebreton, PhD
Editorial Board Member
Communications Psychology
orcid.org/0000-0002-2071-4890

Version 1:

Decision Letter:

Dear Professor Lamm,

Thank you for your patience during the peer-review process. Your manuscript titled "Effort and time costs influence motivational asymmetries in self-benefitting vs pro-environmental decisions" has now been seen by 3 reviewers, and I include their comments at the end of this message. They find your work of interest but raised some important points. We are interested in the possibility of publishing your study in Communications Psychology, but would like to consider your responses to these concerns and assess a revised manuscript before we make a final decision on publication.

We therefore invite you to revise and resubmit your manuscript, along with a point-by-point response to the reviewers. Please highlight all changes in the manuscript text file.

Editorially, we consider important that you further discuss your approach and results, and nuance the interpretations of your findings, so as to take into account the meaningful suggestions of all 3 reviewers. While we appreciate that you are already using Bayesian stats to support your null finding, editorial policy also require that you report Bayes Factors.

I am attaching an Editorial Requests Table that details critical reporting requirements for the revised manuscript. Please attend to each item and ensure your manuscript is fully compliant. If your revised manuscript is not aligned with these requests on major issues, such as those concerning statistics, it may be returned to you for further revisions without re-review.

Please submit the following items:

- Revised manuscript
- Point-by-point response to the referees' comments
- Cover letter (as a separate document)
- <https://www.nature.com/documents/nr-reporting-summary.pdf>>Nature Research Reporting Summary
- Completed Editorial Request Table (attached).

via this link: Link Redacted .

Additional guidance is available in our style and formatting guide <https://www.nature.com/documents/commpsychol-style-formatting-guide-accept.pdf>>Communications Psychology

formatting guide.

Best regards,

Mael Lebreton

Mael Lebreton, PhD
Editorial Board Member
Communications Psychology
orcid.org/0000-0002-2071-4890

REVIEWER EXPERTISE:

Reviewer #1: decision making, pro-environmental /climate change behavior; computational model

Reviewer #2: pro-environmental /climate change behavior

Reviewer #3: decision making; pro-environmental /climate change behavior

REVIEWER REPORTS:

Reviewer #1 (Remarks to the Author):

The authors report experimental and modelling data of effort and time discounting tasks with self-benefitting and pro-environmental outcomes. In line with previous work in the domain of cognitive and social neuroscience, they find discounting effects in both tasks, which are overall more pronounced for the pro-environmental outcomes – which are arguably less attractive than self-benefitting outcomes. Some nuances are observed, for instance that waiting time seems less aversive than physical effort, at least in some individuals. Interestingly, while the differential discounting effect for the different outcome types was not related to self-reported climate change beliefs, it was associated with self-reported support for costly climate policies.

The article addresses an interesting and relevant topic and was a pleasure to read. The methods are clear and appear sound to me (although I am not a modelling expert), the results are presented in a comprehensive manner. I also appreciate the rigorous alignment with the preregistration (and report of deviations thereof), the calibration of effort and time manipulations, as well as the extensive detail in the supplemental material. Most of my comments regard nuances in the interpretation and discussion, integration with the literature, and impact of the findings. My comments are detailed below, in the order of appearance.

Abstract and Introduction:

- The distinction between hyperbolic and parabolic discounting functions is highlighted in the abstract, which is what the models returned. However, in the discussion it is only briefly mentioned that this might indicate different underlying mechanisms, without further elaboration. It would be valuable to elaborate on this difference somewhere in the manuscript or maybe not highlight it in the abstract.
- It might be good to include a more detailed introduction of the physical force measure and the planned analyses and hypotheses in the introduction (unless I missed this). It reads as if the choices are the only relevant dependent measure, while in the results effect on the number and duration of grips is presented.

Methods and results:

- It would be valuable to spell out the approach for the power analysis in the manuscript rather than referring to the preregistration only.
- The task was completed in the lab which probably increases experimenter effects in general and social desirability. A discussion of these influences seems valuable (even if it is not possible to fully control these factors in this type of studies).
- I found it a bit uncommon to start the lab procedure by merely an illustration (Figure 1) without introductory text (the first paragraph immediately describes the force calibration).
- The trial order was pseudorandomized using mini-blocks of 5 trials". This is not mentioned explicitly, but I assume the mini-block structure refers to the outcome type (5 self trials followed by 5 CO2 trials), while effort/reward levels vary within and across these according to a pseudorandomized scheme to maintain average effort levels.
- The mini-block structure of self vs CO2 trials (if I understood this correctly) could promote differences in effort regulation which might not occur when outcome types would be randomly intermixed. Specifically, it has been shown that individuals

would strategically upregulate cognitive control when it is worth it both in terms of outcome magnitude, but also in terms of number of consecutive trials (Kukkonen et al. 2025 <https://journalofcognition.org/articles/10.5334/joc.415>). Here this could mean that the self trials have a larger benefit from such upregulation which might partly explain stronger effort discounting in the CO2 trials. This is not to say that the results are not interesting or valid, but the differential effect might be weaker in an entirely random trial-by-trial manipulation of outcome type.

- Comparing model parameters across tasks and individuals in the computational modeling results the discounting parameters are correlated across tasks and individuals. I found this surprising considering that in the choice data there was a ceiling effect in almost half of the participants in the time task (line 401). Were these participants included in the parameter comparison (line 497)? But it is entirely possible that I am not fully understanding the models.
- Interpersonal differences (line 534): albeit exploratory, it would be valuable to indicate which of these correlations would survive correction for multiple comparisons (Bonferroni?)

Discussion:

- The discussion on the lack of a relationship with climate change beliefs falls a bit short (lines 613-620). Reference is made to previous research, but potential reasons for this lack (beyond the low variability in the self-report data) are not discussed. Here two relevant papers for this discussion: Deltomme et al. 2023 (<https://doi.org/10.3390/su151914484>) and Lange 2025 (<https://doi.org/10.1016/j.jenvp.2024.102381>)

- More generally, it would be valuable to discuss potential reason for the presence and absence of relationships between the experimental data and the self-report data. In addition to lacking individual variability (and potentially also general power considerations) the discrepancy in predictive power between climate change beliefs and policy support could be indicative of differential sources or motivations that underlie these self-report data (e.g., support of costly policies might relate more to the behavioral data since it touches on actual costs for the individual while climate change beliefs are easily adopted and entirely "for free"). The correlation with the reported policy support might reflect alignment of the costs as one crucial factor that individuals consider when making climate-relevant choices (in contrast, in the climate change belief data, there is no direct cost association).

- With respect to low variation as potential reason for the absent relationship with climate change beliefs (line 618) it would be interesting to mention whether the policy support data had a similar "problem" as the climate change belief data (i.e., being too homogeneous).

- Sample size/power could be discussed as common reason for the partial absence of hypothesized relationships with self-report data.

- Temporal discounting (line 601): while certainly true, is unclear why it would be relevant that this effect is observed "across species" - this is the only place in the paper where other species are mentioned. This seems unnecessary, and effort discounting is similarly inherent to many species.

- Parabolic/hyperbolic discounting functions (l. 603): when alluding to the differential underlying cognitive mechanisms it would be valuable to briefly sketch those. As it stands now there is no functional reason given for why these different shapes would occur – and this difference is highlighted in the abstract.

- Ceiling effect of time manipulation: a relevant difference between physical effort and waiting time might be that waiting time in the lab context is less aversive than individual decisions that involve a time cost in real life. The participants already anticipate and commit to spending a certain time in the lab and receive a base payment. In this context some minutes more or less would not make a huge difference.

- I am missing a discussion on real life implications. Indeed, as discussed, the paradigm provides insights into the decision making process with real consequences. The data confirms that pro-environmental outcomes are inferior to self-benefitting ones - which mirrors real-life decisions. That said, how does this data help to change this problem, which is inherent to real life features of these outcomes? Even if equated in monetary value, self-benefitting outcomes are always more immediate, more tangible, closer in time, easier to assess and use. A CO2 reduction has no concrete "value" for the person and it is hence not surprising at all that it is inferior in the lab and in real life. And with this in mind, what factors (in real life) would potentially lead individuals to accept the associated costs (effort or time) when inherent features of the outcome cannot be changed? (far away in time and space, not concrete, ...).

Sincerely,
Ruth Krebs

Reviewer #2 (Remarks to the Author):

Studying pro-environmental behavior via surveys has limited value, since participants can fill in socially desirable responses and there are no real tradeoffs involved in decision making. Following the theory of planned behavior, there might be many intentions of doing the "right" thing, but actual behavioral change does not just relate on intentions.

As such, I appreciate this manuscript by including real choices with physical effort or waiting times in deciding whether to choose for oneself or for the environmental option. The key puzzle of the manuscript is what is actually measured with the environmental option. It is unclear whether participants perceive the donation to an organization as a pro-environmental choice (there is a lot of greenwashing), and I was not surprised by the lack of correlation with the climate change attitudes. You may have received the same findings if the donation was for an organization that donates food or medicine to the Global South. It is no surprise the experiment is based on studies of prosociality, which is the framing here of the environmental issue. This might be the framing that some people see, but it ignores other motivations. Being vegetarian or taking the train does not have perceived as a sacrifice for the environment, but a desired option. It also suggests that reducing CO2 emissions is a socially beneficial activity, while in some countries, like the USA, the federal government actively blocks citizens from doing the "right" thing. There might at least some reflection in the introduction on different aspects influencing

pro-environmental behavior, and that it is not just a tradeoff between self and the environment.

In general the experimental design is clever and well worked out, the analysis is sophisticated and the results are in line with my expectations. As mentioned above, the main challenge is understanding what we are actually measuring with the pro-environmental option.

More detailed comments:

The paper mentioned that in the environmental treatment a win is “translated into money that would be donated to an organisation that actively combats climate change by reducing greenhouse gas emissions through projects that promote renewable energy sources, energy-efficient technologies, and sustainable solutions”. It would be helpful if information was provided what was communicated. At least it would be helpful to reflect on whether this is actually seen as a real contribution to reduce CO₂ emissions since it depends on the trust participants have on the impact of the investment. Giving money to an organization is not really a pro-environmental action, and donations are used a lot in business as green washing.

Figure 1 is not clear. I would have expected a decision tree. It would have been appropriate if the action procedure was in the supplemental material.

Check the references both in the main text and in the supplemental material. Not all cited pubs are in the references.

Reviewer #3 (Remarks to the Author):

Review of the Manuscript

The paper presents a preregistered experimental study investigating how reward type and magnitude influence individuals' willingness to engage in tasks that vary in required effort and time. Specifically, the authors examine whether rewards targeted at the self versus the environment affect motivation, and they analyze discounting curves (reward vs. effort or time), linking estimated parameters to external variables such as climate change beliefs and policy support.

Overall, I am very positive about this paper for several reasons. It is well-written, clearly motivated, statistically well-powered, and preregistered (with accurate justifications in the case of deviations) Moreover, it introduces a potentially powerful behavioral paradigm for studying real-world pro-environmental behavior. The results are compelling—including the null findings, which are informative given the high statistical power. The paradigm is convincing in its ability to model trade-offs that resemble those encountered in everyday life. I particularly appreciate the attention given to potential differences between time and effort discounting. However, I do feel the paper would benefit from a dedicated section discussing limitations.

Disclaimer: I do not have extensive expertise in Bayesian statistics. While I understood the analyses as presented, I am not in a position to evaluate their technical rigor or completeness.

My comments are intended to help improve the manuscript and/or inspire further reflection. I believe the paper is worthy of publication.

Specific Comments and Suggestions

- Policy Support Measure: As this is a newly developed instrument, it may benefit from further elaboration. Given the study's focus on personal versus collective gain, it could be informative to explore—post hoc—whether items referencing personal versus societal costs differentially predict willingness to work or wait for personal versus environmental rewards. A factor analysis (FA) or principal component analysis (PCA), perhaps included in the appendix, could help clarify the internal structure of the measure.

- Line 194 (“participants were given no information”): This phrasing is ambiguous. I assume it means that specific numerical information was withheld. However, was ordinal information provided, or were participants expected to infer levels through sensory cues? Including the exact instructions given to participants would help clarify this point.

- Time-Effort Calibration: I found this aspect of the design intriguing. The supplementary information suggests it was challenging to implement. While the rationale is sound, it may be worth revisiting this in the discussion section to identify potential limitations. For instance, differing discounting curves could inherently compromise the accuracy of indifference point identification. Conversely, the reliability of curve estimation may depend on the precision of those indifference points, which appeared problematic for some participants. This is not necessarily a flaw—your methodological care is commendable—but it would be helpful to acknowledge interpretive challenges and suggest avenues for future research. Many real-world activities (e.g., protesting) involve both time and effort. Moreover, is time perceived similarly across low and high effort conditions? Extremely low-effort tasks may be experienced as boring, which could influence discounting. Briefly flagging this as a topic for future exploration would be valuable.

- Lines 601–602: I am not fully convinced that the literature on future-oriented time discounting—despite its robust evidence for hyperbolic discounting—can be directly applied to discounting in the present context. Why should current time discounting be analogous to future time discounting? It would strengthen the argument to include a line clarifying this assumption or to nuance the comparison.

- Contingency Design Choice: A notable feature of your design is the contingency applied when participants fail to follow through on their chosen plan—either by exerting insufficient effort or prematurely ending the waiting period. In such cases, they receive no reward. While this is methodologically defensible, one could argue that participants demonstrated good intentions and partial commitment, which might justify awarding them the default reward. From a fairness perspective, this could be seen as more balanced. Did you observe any participant reactions to this aspect of the design? Although not essential for this paper, a follow-up analysis could explore potential behavioral consequences. In real-world pro-environmental actions, individuals may attempt to act sustainably (e.g., cycling instead of driving) but encounter unforeseen obstacles (e.g., a flat tire or traffic jam). Such setbacks may lead to feelings of discouragement or perceptions of unappreciated effort. Your design may inadvertently simulate this dynamic. In effect, participants are choosing between a small, guaranteed reward and a larger, uncertain one—with a 90% chance of success or a 10% chance of receiving nothing. While this risk may be under their control in the case of waiting time, I question whether that holds true for effort-based tasks, especially at higher levels. It would be helpful to briefly justify this design choice and/or acknowledge it as a limitation or avenue for future research.
- Line 311: Consider adding the term “reward” to clarify that the discounting pertains specifically to reward valuation. Is this interpretation correct?
- Line 471: Please review this sentence. It appears that “individual difference” may have been used in place of “environmental reward” or a similar construct.
- Line 340: The numeral “1” before the fraction seems out of place and may be a typographical error. If it does not convey meaningful information, consider removing it.
- Line 484: The statement here reads as somewhat self-evident: if the standard deviation of variable 1 exceeds that of variable 2, then rank order reversals are expected. I believe the intended point is that, despite the overall main effect, a substantial subset of participants found the environmental reward more motivating. If so, rephrasing to emphasize this nuance would enhance clarity.
- Concluding Note: As behavioral researchers, we often emphasize relative differences—such as the greater effectiveness of self-rewards over pro-environmental ones. However, it is striking that environmental rewards still elicit substantial effort and time investment in absolute terms. It would be worthwhile to highlight this somewhere in the manuscript, perhaps by quantifying the average amount of effort exerted or time spent “for the environment” per participant.
Best of luck with the revision!

* TRANSPARENT PEER REVIEW: Communications Psychology uses a transparent peer review system. This means that we publish the editorial decision letters including Reviewers' comments to the authors and the author rebuttal letters online as a supplementary peer review file. However, on author request, confidential information and data can be removed from the published reviewer reports and rebuttal letters prior to publication. If your manuscript has been previously reviewed at another journal, those Reviewers' comments would not form part of the published peer review file.

If you experience problems in linking your ORCID, please contact the Platform Support Helpdesk.

Version 2:

Decision Letter:

Dear Professor Lamm,

Your manuscript titled "Effort and time costs influence motivational asymmetries in self-benefitting vs pro-environmental decisions" has now been seen by our reviewers, whose comments appear below. In light of their advice I am delighted to say that we are happy, in principle, to publish a suitably revised version in Communications Psychology.

We therefore invite you to revise your paper one last time to address the remaining concerns of our reviewers and a list of editorial requests. At the same time we ask that you edit your manuscript to comply with our format requirements and to maximise the accessibility and therefore the impact of your work.

EDITORIAL REQUESTS:

SUBMISSION INFORMATION:

OPEN ACCESS:

* DATA AVAILABILITY:

Link Redacted

Best regards,

Jennifer Bellingtier

Jennifer Bellingtier, PhD

Senior Editor
Communications Psychology

Mael Lebreton, PhD
Editorial Board Member
Communications Psychology
orcid.org/0000-0002-2071-4890

REVIEWER EXPERTISE:

Reviewer #1: decision making, pro-environmental /climate change behavior; computational model

Reviewer #2: pro-environmental /climate change behavior

Reviewer #3: decision making; pro-environmental /climate change behavior

REVIEWERS' COMMENTS:

Reviewer #1 (Remarks to the Author):

The authors have provided a thorough revision, further strengthening the manuscript. This is a well-deigned study contributing to the growing body of experimental approaches to understand pro-environmental decisions. I have no further comments and would recommend publication.

Reviewer #2 (Remarks to the Author):

My comments have been properly addressed. thx!

Reviewer #3 (Remarks to the Author):

I want to thank the authors for being so responsive. the additions make the paper clearer and stronger. I support the publication of this paper.

Effort and time costs influence motivational asymmetries in self-benefitting vs pro-environmental decisions

Response to reviewers

On behalf of all of the co-authors, we would like to thank the editorial board and the reviewers for the time and effort they put into reviewing our manuscript. We believe that your constructive comments have greatly benefitted the paper, and hope it now fully fulfills the standards of publication in Communications Psychology. In brief, we have expanded the methods section and adjusted our figures for clarity, substantially extended the discussion, ran additional analyses exploring the effects of social desirability and also supported all null findings with Bayesian statistics (as per the editorial policy). We have uploaded two manuscript documents, a clean one and one with the respective changes highlighted. In the following, we provide a point-by-point reply to all comments.

Reviewer #1 (Remarks to the Author):

The authors report experimental and modelling data of effort and time discounting tasks with self-benefitting and pro-environmental outcomes. In line with previous work in the domain of cognitive and social neuroscience, they find discounting effects in both tasks, which are overall more pronounced for the pro-environmental outcomes – which are arguably less attractive than self-benefitting outcomes. Some nuances are observed, for instance that waiting time seems less aversive than physical effort, at least in some individuals. Interestingly, while the differential discounting effect for the different outcome types was not related to self-reported climate change beliefs, it was associated with self-reported support for costly climate policies.

The article addresses an interesting and relevant topic and was a pleasure to read. The methods are clear and appear sound to me (although I am not a modelling expert), the results are presented in a comprehensive manner. I also appreciate the rigorous alignment with the preregistration (and report of deviations thereof), the calibration of effort and time manipulations, as well as the extensive detail in the supplemental material. Most of my comments regard nuances in the interpretation and discussion, integration with the literature, and impact of the findings. My comments are detailed below, in the order of appearance.

We thank Dr. Krebs for the thoughtful review and positive evaluation of our work!

Abstract and Introduction:

- The distinction between hyperbolic and parabolic discounting functions is highlighted in the abstract, which is what the models returned. However, in the discussion it is only briefly mentioned that this might indicate different underlying mechanisms, without further elaboration. It would be valuable to elaborate on this difference somewhere in the manuscript or maybe not highlight it in the abstract.

Thank you for the helpful suggestion. We have now expanded the discussion section to include a more elaborate explanation of the different processes involved in discounting effort vs. time.

Page 20. *“Our computational modelling analyses showed distinct discounting patterns for effort and time. Effort discounting followed a parabolic function, consistent with prior research on effort-based decision-making^{14,16,17,19,27}, whereas temporal discounting was best captured by a hyperbolic function, a well-established characteristic of temporal decision-making¹². A parabolic form implies that the subjective devaluation of rewards grows disproportionately as effort demands increase – the subjective value drops slowly when the required level of effort is low but accelerates when the effort required gets higher. By contrast, a hyperbolic form implies that the subjective value drops sharply even for short waiting times, then flattens as delays lengthen, consistent with the immediacy effect and diminishing marginal opportunity costs of additional delay. Although experiential delays in the range of seconds, as used in our paradigm, differ from the longer horizons more commonly studied in the temporal discounting research, both literatures converge on hyperbolic discounting as the characteristic functional form, suggesting continuity across timescales^{12,14,59,60}. Taken together, these functional differences suggest that effort and time costs are processed through distinct cognitive and physiological mechanisms. This aligns well with evidence from neuroimaging findings reporting the engagement of distinct brain areas for each type of cost⁹, and highlights the need to study these costs separately in future decision-making research.*

- It might be good to include a more detailed introduction of the physical force measure and the planned analyses and hypotheses in the introduction (unless I missed this). It reads as if the choices are the only relevant dependent measure, while in the results effect on the number and duration of grips is presented.

We agree with the reviewer that our description was not clear enough. Given that choices, is our main variable of interest for the majority of the analyses we report (we have linear mixed models with choice as DV for both tasks, and choice is the DV for the computational modelling), there is a particular focus on that in the introduction. But indeed, one of our preregistered analysis was focused on the force data from the effort task (more concretely on the amount of force exerted over the 3 second period, measured as area under the curve) and that was not described clearly in the introduction. As the second analysis we did on the force data (i.e., the amount of time participants pressed with the required degree of force) was not preregistered and is marked as exploratory, we have not included it in the introduction. We have now rewritten and expanded this section to be more explicit about our preregistered hypotheses and expectations in the Introduction, which match the relevant parts in the Results section:

Page 4. *“Our key preregistered hypotheses were threefold. First, we expected that participants would be more likely to select the costly option for themselves compared to the environment in both tasks. In line with research on prosociality, for the effort task, we further hypothesized that when participants did choose the effortful option for the environment, they would energize their actions to a lesser extent than when exerting effort for themselves, as reflected in pressing with less force on the grip-force device, indicating a discrepancy between decisions and actual behavior. Second, we anticipated that this reduced willingness to select the costly option for the environment in both tasks would be captured in the computational modeling as stronger discounting for environmental*

relative to self-benefits. Finally, we expected that the difference in discounting scores for self versus environmental outcomes would be associated with participants' climate change beliefs and their willingness to support costly climate policies.

Methods and results:

- It would be valuable to spell out the approach for the power analysis in the manuscript rather than referring to the preregistration only.

Thank you for this comment. The power analysis is described in detail in the supplement (Section 2.1.), and we need to refrain from moving it to the main text due to the journal's space limits. We briefly summarized its main aspects in the main manuscript as follows:

Page 4. *"The targeted sample size was based on a power analysis focusing on the expected difference between self and environmental choices and is described in detail in the SI, Section 2.1. and in our preregistration (<https://osf.io/bksf9>, preregistration date: 16.03.2022)."*

- The task was completed in the lab which probably increases experimenter effects in general and social desirability. A discussion of these influences seems valuable (even if it is not possible to fully control these factors in this type of studies).

Thank you for raising this important point. We took steps to mitigate these effects, e.g., we placed the participants in a separate room from the experimenter so that the experimenter could not see their monitor or decisions. Additionally, we still assessed whether the setting led to them feeling observed – we asked an open-ended question at the end of the experiment about whether they felt observed at any time during the experiment. Out of 74 people, only 11 wrote anything other than "no". Only two out of these 11 wrote "yes", and the rest wrote things such as "unsure", "maybe a little at the beginning", or similar statements.

To ensure that this aspect did not bias our results, we re-ran the analyses excluding all participants who wrote anything other than "no" to that question ($n = 11$). The pattern of results remained largely unchanged, with the only difference that the recipient main effect became significant (from $p = 0.069$ in the full sample to $p = 0.031$) in the time task, indicating lower willingness to select the costly option when rewards benefited the environment. We have now included these results in the supplemental information file (Section 3.4.3).

Nevertheless, it remains possible that merely being in a formal lab setting increases demand effects. And indeed, in the reviewer's work, some studies have shown that the self-environment difference was larger in an online setting compared to a lab setting (Krebs et al., 2023). We thus briefly discuss this aspect now in the limitations section.

Page 12. *"To explore potential social desirability effects, we also ran the models excluding participants who wrote anything other than "no" in the open-ended question asking whether they felt observed when making their choices (excluded $n = 11$). The results remained largely the same, with the only notable difference being that the effect of the recipient became significant in the time task (see SI Section 3.4.3 for the full output of the models)."*

and

Page 23. *"A potential inherent limitation of our design is that the tasks were completed in the laboratory, which may increase social desirability effects. We attempted to minimise these by placing participants in a separate room from the experiments for the main part of the experiment, and excluding people who wrote anything other than "no" to questions about feeling observed did not change the results meaningfully. However, prior work on pro-environmental motivation has documented differences between lab and online behavior, such that the self-environment differences were larger in the online setting²⁸, so a potential future direction could be to adapt these paradigms for online use, or manipulate social desirability, and assess whether the magnitude of the effects will differ."*

Krebs, R. M., Prevel, A., Hall, J. M., & Hoofs, V. (2023). Think green: Investing cognitive effort for a pro-environmental cause. *Journal of Environmental Psychology*, 85, 101946.

- I found it a bit uncommon to start the lab procedure by merely an illustration (Figure 1) without introductory text (the first paragraph immediately describes the force calibration).

Thank you for sharing this observation. We added an introductory paragraph before describing each step in detail.

Page 5. *"Seventy-four participants completed the experimental part of the study in a single laboratory session that consisted of several stages. Upon arrival, they were welcomed to the lab, were briefed and received instructions. The session began with preparatory steps, including calibration procedures and familiarisation with the effort and time levels, as well as several practice trials to ensure that participants understood the task procedure. The experimental tasks followed, where participants made repeated decisions involving effort and waiting costs for self-benefiting and pro-environmental outcomes. An overview of the procedure is shown in Figure 1, and each stage is described in detail below."*

- The trial order was pseudorandomized using mini-blocks of 5 trials". This is not mentioned explicitly, but I assume the mini-block structure refers to the outcome type (5 self trials followed by 5 CO2 trials), while effort/reward levels vary within and across these according to a pseudorandomized scheme to maintain average effort levels.

We appreciate the opportunity to clarify. We constructed the mini-blocks of 5 trials with two considerations in mind. First, we ensured that the average effort level within each mini-block was between levels 2 and 3, which prevented participants from encountering sequences of only very high-effort trials and thus reduced potential fatigue effects. Second, the trial order was pseudorandomized such that the same recipient type (self vs. environment) did not appear more than 4 times consecutively. Together, these constraints guided the pseudorandomization and ensured both variability of effort levels and balance across recipient conditions.

We rewrote the description of the randomization to enhance clarity:

Page 8. *"The trial order was pseudorandomised using mini-blocks of five trials, maintaining an average effort level of 2–3 over the previous five trials to avoid fatigue. The pseudorandomization also ensured that the same recipient did not appear more than four times in a row (i.e., participants were never exposed to five or more self trials in a row)."*

-The mini-block structure of self vs CO₂ trials (if I understood this correctly) could promote differences in effort regulation which might not occur when outcome types would be randomly intermixed. Specifically, it has been shown that individuals would strategically upregulate cognitive control when it is worth it both in terms of outcome magnitude, but also in terms of number of consecutive trials (Kukkonen et al. 2025 <https://journalofcognition.org/articles/10.5334/joc.415>). Here this could mean that the self trials have a larger benefit from such upregulation which might partly explain stronger effort discounting in the CO₂ trials. This is not to say that the results are not interesting or valid, but the differential effect might be weaker in an entirely random trial-by-trial manipulation of outcome type.

We thank the reviewer for raising this interesting point and for drawing our attention to Kukkonen et al. (2025). However, our design did not involve fixed mini-blocks of recipient type (i.e., 5 self trials followed by 5 CO₂ trials). Rather, as mentioned above, recipient type was pseudorandomized such that it varied across trials, with the additional constraint that the same recipient did not appear more than 4 times consecutively. Thus, participants did not encounter extended blocks of self vs. CO₂ trials that could systematically promote differential effort regulation. We agree that future work could explicitly test whether strategic upregulation differs between trial-by-trial randomization and longer blocks of recipient type.

- Comparing model parameters across tasks and individuals in the computational modeling results the discounting parameters are correlated across tasks and individuals. I found this surprising considering that in the choice data there was a ceiling effect in almost half of the participants in the time task (line 401). Were these participants included in the parameter comparison (line 497)? But it is entirely possible that I am not fully understanding the models.

We appreciate the reviewer's observation regarding the correlation of discounting parameters across tasks. To clarify, all participants (including those exhibiting a ceiling effect) were included in the parameter comparisons. We used computational models to estimate latent parameters (e.g., k) that reflect underlying decision processes, even when observed choices are constrained. This approach allows us to capture meaningful individual differences in motivation and cost sensitivity, even if some participants consistently chose the pro-environmental option in the time task. The observed correlations align with our preregistered expectation: *"However, if people are pro-environmental in one task (e.g., the effort task), we suspect they will be pro-environmental in the other (e.g., the time task), aiming to maximize rewards in both cases."*

Specifically, we found that individuals who were self-motivated in one task tended to be self-motivated in the other (a positive correlation between k effort self and k time self). We also found that those willing to incur costs for pro-environmental outcomes in one task were more likely to do so in the other (a positive correlation between k effort env and k time env). Most importantly, the relative motivation for self vs environment, as captured by the k diff, showed the strongest correlation between tasks ($\tau = 0.57$, $p < 0.001$). This suggests that both tasks measure a common underlying construct: the degree to which individuals prioritize environmental outcomes over self-benefiting ones, despite associated costs.

To improve clarity, we revised the text presenting these results as follows:

Page 17. *“Importantly, the differences between the discounting parameters for self and the environment (k effort diff and k time diff) were also significantly correlated ($r = 0.57$, $p < 0.001$). This indicates that participants who were more willing to incur effort costs for pro-environmental outcomes (relative to self outcomes) were also more willing to incur time costs for pro-environmental outcomes (relative to self outcomes).”*

Regarding the concern about ceiling effects in the time task: Participants who displayed such patterns were still included in the computational modelling, as hierarchical Bayesian models—the approach we used in our study—are particularly well-suited to accommodate limited variability in observed choice data, including patterns such as ceiling effects. These models estimate discounting parameters that best explain the available responses (i.e., if there is limited variability, the discounting factor k would be close to 0). The resulting parameters therefore still reflect interindividual differences, even if some participants’ raw choice distributions were restricted. In such cases, the values will be close to 0 but would still capture the (rather small) variability in their responses.

- Interpersonal differences (line 534): albeit exploratory, it would be valuable to indicate which of these correlations would survive correction for multiple comparisons (Bonferroni?)

We thank the reviewer for this suggestion. We agree that multiple-comparison corrections play an important role in confirmatory hypothesis testing. However, we consider their application less appropriate in the present case, for two reasons. First, the interpersonal correlations were explicitly framed as exploratory, rather than as formal hypothesis tests. The purpose of these analyses is to provide a descriptive overview of potential relationships that may inspire future work, rather than to establish definitive claims. Second, our study was powered to detect differences in discounting between self and environment conditions, not to detect small interpersonal correlations. Applying a highly conservative correction such as Bonferroni in this exploratory context would disproportionately increase the risk of Type II error, potentially masking genuine and theoretically meaningful associations. To balance transparency and caution, we have highlighted these results as exploratory and interpret them accordingly, without drawing strong inferential conclusions. We reviewed the text and ensured these findings are labeled accurately as exploratory and interpretation is cautious. We also added a sentence to the results section emphasizing that the results are uncorrected and warrant caution with interpretation.

Page 18. *“Observing substantial inter-individual differences in discounting prompted us to explore the relationship between interpersonal differences in self-report measures and the differences in discounting tendencies. As these interpersonal correlations were examined in an exploratory manner, results are reported uncorrected for multiple comparisons and should be interpreted with appropriate caution.”*

Discussion:

- The discussion on the lack of a relationship with climate change beliefs falls a bit short (lines 613-620). Reference is made to previous research, but potential reasons for this lack (beyond the low variability in the self-report data) are not discussed. Here two relevant papers for this discussion: Deltomme et al. 2023 (<https://doi.org/10.3390/su151914484>) and Lange 2025 (<https://doi.org/10.1016/j.jenvp.2024.102381>)

We thank the reviewer for this remark. In the revised manuscript we expanded the beliefs section to offer additional explanations for the null association, beyond limited variability in the self-report data. Specifically, we now additionally 1) discuss the attitude–behavior gap, emphasizing that belief endorsement can be largely symbolic and low-cost whereas our tasks require cost-incurring action; 2) acknowledge power considerations given that the study was designed primarily for self–environment contrasts rather than small belief–behavior effects; and 3) report complementary Bayesian analyses quantifying the strength of evidence for the null. We believe this expanded discussion provides a clearer and more comprehensive rationale for the lack of significant association between beliefs and the discounting difference.

We appreciate the two suggested papers (Deltomme et al., 2023; Lange, 2025) and agree that they raise important points about how *pro-environmental behavior* is measured. However, we feel they are not directly applicable to our specific issue, which concerns climate-change beliefs (which is an attitudinal construct rather than a behavioral measure). While we do agree that beliefs can be seen as potentially more tightly related to the person-property measures of pro-environmental behavior Lange talks about, we are cautious about treating beliefs as interchangeable with behavior measures in this context. Thus, we opted not to anchor our beliefs discussion in those measurement papers, but instead expanded the section to articulate why beliefs may fail to predict costly action in our paradigm, while pointing readers to predictors more tightly linked to costly behavior (e.g., policy support).

Page 21-22. *“Contrary to our preregistered expectations, climate change beliefs were no significant predictors of the difference in discounting parameters. Bayesian analyses largely supported these null effects, providing moderate to strong evidence for the null hypothesis for both tasks, with the exception of climate change belief certainty in the effort task, where the evidence for the null hypothesis was less strong but still favouring it. There are several potential explanations for the overall lack of associations. First, there is a growing body of research showing that attitudes do not necessarily translate to action, especially when costs are high^{53,54}. Beliefs can be endorsed at little or no personal cost. For instance, agreeing with the view that climate change is a serious problem can be largely symbolic and will not require individuals to act upon this view or to sacrifice resources. As a result, these beliefs may capture general attitudes rather than the motivational processes that guide cost-incurring decisions. Second, it is important to note, that our sample reported overall high climate change beliefs, with 77% of participants selecting the highest degree of certainty that climate change is happening (see SI Fig S3). The related lack of interindividual variation may thus be an alternative explanation of why we did not find a significant relationship between beliefs and behaviour. That said, other research has shown little difference between sceptics and believers in terms of engaging in costly actions⁶¹. Relatedly, our power analysis was primarily designed to detect self-environment differences in discounting; thus, if the true association between climate change beliefs and costly behavior is very small, the study may have been underpowered to reliably detect it. This notwithstanding, our study supports prior findings suggesting that climate change beliefs, while important for signaling general attitudes, do not necessarily translate into actual willingness to incur personal costs. This underscores the value of focusing on predictors more tightly linked to costly pro-environmental behavior.*

By contrast, we confirmed our preregistered hypothesis that participants who exhibited higher discounting of environmental rewards compared to self-benefitting rewards were also less likely to support costly climate policies, as measured with a novel scale we developed to

focus on trade-offs in policies. One reason may be that policy support inherently entails an acceptance of potential personal or societal costs, making it more closely aligned with cost-sensitive choices in our paradigm.”

We also want to note that in line with the journal’s editorial policy, we now conducted Bayesian analyses to accompany all null findings (including the belief measures). As part of implementing these analyses, we also performed a thorough code review. This confirmed that the models were specified correctly, with one minor exception in a single model for the effort task—after correcting this, the p-value for climate-change belief certainty in the effort task changed slightly (from 0.096 to 0.052), thus, remaining non-significant and not altering our conclusions. Importantly, the Bayesian results were consistent with the frequentist analysis: we found moderate to strong evidence for the null across predictors, with the exception of belief certainty in the effort task, where the evidence for the null was anecdotal (i.e., still favoring H_0 over H_1 , but not as strong as for the other predictors). The revised Discussion reflects these updates, and the Supplemental Materials (Section 2.7.) document the Bayesian procedures in full. We believe the new discussion points regarding the lack of a statistically significant association provide a clear and balanced account of the finding by e.g., quantifying the strength of evidence, situating it within the attitude-behavior literature with a particular focus on cost, and acknowledging the potentially limited power for very small effects.

- More generally, it would be valuable to discuss potential reason for the presence and absence of relationships between the experimental data and the self-report data. In addition to lacking individual variability (and potentially also general power considerations) the discrepancy in predictive power between climate change beliefs and policy support could be indicative of differential sources or motivations that underlie these self-report data (e.g., support of costly policies might relate more to the behavioral data since it touches on actual costs for the individual while climate change beliefs are easily adopted and entirely "for free"). The correlation with the reported policy support might reflect alignment of the costs as one crucial factor that individuals consider when making climate-relevant choices (in contrast, in the climate change belief data, there is no direct cost association).

We thank you for this thoughtful suggestion. We agree that it is valuable to reflect on why certain self-report measures relate to the experimental data while others do not. As you note, one explanation is that climate change beliefs can be endorsed at little or no personal cost and may therefore show weaker associations with cost-incurring behaviors, whereas support for costly climate policies directly engages considerations of individual cost and trade-offs. We added a discussion of these points, highlighting that differences in predictive power between belief- and policy-related measures may reflect distinct underlying motivations and processes.

“...Another reason might be that beliefs can be endorsed at little or no personal cost. For instance, agreeing with the view that climate change is a serious problem can be largely symbolic and will not require individuals to act upon this view or to sacrifice resources. As a result, these beliefs may capture general attitudes rather than the motivational processes that guide cost-incurring decisions.”

Page 21. *“By contrast, we confirmed our preregistered hypothesis that participants who exhibited higher discounting of environmental rewards compared to self-benefiting rewards were also less*

likely to support costly climate policies, as measured with a novel scale we developed to focus on trade-offs in policies. One reason may be that policy support inherently entails an acceptance of potential personal or societal costs, making it more closely aligned with cost-sensitive choices in our paradigm."

- With respect to low variation as potential reason for the absent relationship with climate change beliefs (line 618) it would be interesting to mention whether the policy support data had a similar "problem" as the climate change belief data (i.e., being too homogeneous).

The policy support data displayed a more normal distribution than, e.g., climate change belief certainty, albeit it was still right-skewed. We have included a plot of the distribution of the policy support measure in Supplementary Figure S2B and added the following text to describe that.

SI, Page 8. Regarding the distribution of the climate change key beliefs ratings, it is important to note that we lacked participants with the lowest levels of belief. There were no participants who rated their climate change certainty as lower than 5 on a 1-9 scale, and no participants who reported thinking climate change is not happening or caused mainly by natural causes. The distribution of the policy support measure exhibited a right-skew (mean = 4.18 on a scale from 1 to 5), but the responses were more evenly distributed compared to the majority of the climate change belief subscales."

- Sample size/power could be discussed as common reason for the partial absence of hypothesized relationships with self-report data.

Thank you for this comment, we added a sentence about this in the section where we describe the lack of relationship between climate change beliefs and discounting tendencies.

Page 21. Relatedly, our study was primarily powered to detect differences in discounting between self and environment conditions; thus, if the true association between climate change beliefs and costly behavior is very small, we may have been underpowered to detect it reliably.

- Temporal discounting (line 601): while certainly true, is unclear why it would be relevant that this effect is observed "across species" - this is the only place in the paper where other species are mentioned. This seems unnecessary, and effort discounting is similarly inherent to many species.

We removed the species argument from the discussion.

- Parabolic/hyperbolic discounting functions (l. 603): when alluding to the differential underlying cognitive mechanisms it would be valuable to briefly sketch those. As it stands now there is no functional reason given for why these different shapes would occur – and this difference is highlighted in the abstract.

We thank you for pointing this out. Temporal discounting is typically well captured by a hyperbolic function, which has been argued to reflect the strong immediacy effect (steep devaluation of short delays) followed by relatively shallower decreases at longer delays, consistent with the role of opportunity costs. In contrast, effort discounting is better fit by a

parabolic function, which implies that subjective costs increase disproportionately at higher effort levels. This convex pattern aligns with both physiological and psychological evidence that perceived effort increases nonlinearly as demand intensifies. In response to this and the first point of your review, we have added a short discussion of these distinct functional forms to clarify that the divergence likely reflects different underlying cognitive and physiological mechanisms.

Page 20. *“Our computational modelling analyses showed distinct discounting patterns for effort and time. Effort discounting followed a parabolic function, consistent with prior research on effort-based decision-making^{14,16,17,19,27}, whereas temporal discounting was best captured by a hyperbolic function, a well-established characteristic of temporal decision-making¹². A parabolic form implies that the subjective devaluation of rewards grows disproportionately as effort demands increase, in line with physiological and psychological evidence. By contrast, hyperbolic temporal discounting reflects steep early devaluation of delayed rewards that then flattens, consistent with the immediacy effect and with the role of opportunity costs. These functional differences suggest that effort and time costs are processed through distinct cognitive and physiological mechanisms, as further evidenced by neuroimaging findings reporting the engagement of distinct brain areas for each type of cost⁹, reinforcing the need to study these costs separately in future decision-making research.”*

- Ceiling effect of time manipulation: a relevant difference between physical effort and waiting time might be that waiting time in the lab context is less aversive than individual decisions that involve a time cost in real life. The participants already anticipate and commit to spending a certain time in the lab and receive a base payment. In this context some minutes more or less would not make a huge difference.

We appreciate the reviewer’s point. We informed participants that the duration of the experiment would depend on their choices, which was intended to make the time costs salient. At the same time, we acknowledge that by agreeing to participate in a laboratory study, participants may already have anticipated spending a substantial amount of time in the lab, which could reduce the subjective impact of time costs. We will add this point as a limitation in the Discussion.

Page 23. *“Although participants were told that the duration of the experiment depended on their choices, it remains possible that by committing to a laboratory session, they already anticipated spending a considerable amount of time in the lab, which may have reduced the subjective salience of time costs.”*

- I am missing a discussion on real life implications. Indeed, as discussed, the paradigm provides insights into the decision making process with real consequences. The data confirms that pro-environmental outcomes are inferior to self-benefitting ones - which mirrors real-life decisions. That said, how does this data help to change this problem, which is inherent to real life features of these outcomes? Even if equated in monetary value, self-benefitting outcomes are always more immediate, more tangible, closer in time, easier to assess and use. A CO2 reduction has no concrete "value" for the person and it is hence not surprising at all that it is inferior in the lab and in real life. And with this in mind, what factors (in real life) would

potentially lead individuals to accept the associated costs (effort or time) when inherent features of the outcome cannot be changed? (far away in time and space, not concrete, ...).

We appreciate this thoughtful comment. We were initially hesitant to make strong claims about real-life implications, given the laboratory nature of our paradigm. Additionally, our current data do not directly help solve the structural problem that pro-environmental outcomes are inherently less immediate, tangible, and personally valuable than self-benefitting ones. We see the main value of this data as a starting point to allow others to use this paradigm as a tool for studying ways to address this challenge. Specifically, the task would allow for experimentally testing different interventions (e.g., framing, norm cues, incentive structures) under controlled conditions, before implementing costly or large-scale interventions in the real world. More broadly, we believe that people's willingness to accept *non-financial* costs (such as effort or time) has not been sufficiently studied in environmental psychology, and is underrepresented in intervention research. We hope that our paradigm can contribute to filling this gap and inspire future work that develops and evaluates interventions targeting these types of costs.

We have added a brief note in the Discussion outlining potential directions for future research and interventions that could encourage individuals to accept costs for pro-environmental outcomes despite their inherent disadvantages.

Page 22. *“Beyond these applications, our paradigm could also provide a means to experimentally test interventions aimed at increasing people’s willingness to accept non-financial costs such as effort or time for environmental benefit. These types of cost are underrepresented in current intervention research, despite being highly relevant for everyday sustainable behaviors. Using the paradigm to evaluate such strategies in a controlled setting could therefore help identify promising approaches before implementing large-scale interventions in the real world.*

It will, however, first be of central importance to further validate how the laboratory-experimental setup emulates discounting of environmental benefits under everyday life conditions, and how our findings can be generalised to decision-making outside of the lab.”

Additionally, we highlight this point further in the conclusion part of the manuscript:

Page 24. *“While our findings show that behavioural costs act as barriers to higher motivation, the substantial inter-individual variation captured by our paradigm and the fact that some people do engage in costly climate actions indicate that these barriers are not insurmountable. Therefore, pinpointing the factors that lead individuals to accept time and effort costs would be crucial for designing interventions and policies that support sustained engagement in pro-environmental behaviour.”*

Reviewer #2 (Remarks to the Author):

Studying pro-environmental behavior via surveys has limited value, since participants can fill in socially desirable responses and there are no real tradeoffs involved in decision making.

Following the theory of planned behavior, there might be many intensions of doing the "right" thing, but actual behavioral change does not just relate on intensions.

As such, I appreciate this manuscript by including real choices with physical effort or waiting times in deciding whether to choose for oneself or for the environmental option. The key puzzle of the manuscript is what is actually measured with the environmental option. It is unclear whether participants perceive the donation to an organization as a pro-environmental choice (there is a lot of greenwashing), and I was not surprised by the lack of correlation with the climate change attitudes. You may have received the same findings if the donation was for an organization that donates food or medicine to the Global South. It is no surprise the experiment is based on studies of prosociality, which is the framing here of the environmental issue...

We thank the reviewer for raising this important point. We agree that the interpretation of what the "environmental option" represents is crucial for understanding the paradigm. At the start of the experiment, we devoted substantial time to carefully instructing participants about the meaning of the pro-environmental option. In particular, we were explicit that the donations would be directed toward the direct reduction of greenhouse gas emissions, and we deliberately avoided mentioning any secondary or prosocial co-benefits (e.g., creation of jobs, social development). The instructions given to participants highlighted that every point earned for the environmental organization would translate into real monetary donations, supporting projects that directly reduce CO₂ emissions through renewable energy expansion and efficiency measures. In this way, participants were clearly informed that their choices represented contributions to greenhouse gas reductions, rather than broader prosocial causes.

Relating to the question of whether the findings would be the same if we compared a food charity to CO₂ reduction, we ran a study on that (currently available as a preprint: [https://osf.io/preprints/psyarxiv/c8yxv_v1](https://osf.io/preprints/psyarxiv/c8yxv/v1)). In that study, we compared how willing people are to invest effort in benefitting a food charity compared to a climate change mitigation charity and tested how different climate change interventions change people's motivation. In the control condition of the experiment, we saw that people were significantly more likely to decide to put effort into benefitting the food charity compared to the climate change charity. Their motivation to benefit the climate change charity was specifically related to different pro-environmental questionnaire measures, while the motivation to benefit the food charity was not, indicating motivation to help the climate might be partially distinct from motivation to help other causes.

We have now added a section to the discussion detailing how the environmental option was framed and described to participants (SI, Section 2.5.)

SI, Page 7. *"To ensure clarity and credibility, we were careful in framing what the money "won" for the CO₂ reduction initiative meant. Participants were explicitly informed that we had selected a specific organization with a strong track record in implementing effective greenhouse gas reduction measures, so that their contributions would translate into genuine and direct reductions in CO₂ emissions rather than vague or symbolic actions. At the same time, we deliberately did not disclose the name of the organization to participants. This was done to minimize the risk that prior knowledge, experiences, or attitudes toward a specific organization would influence decisions. Below, we have included the exact description provided to the participants during the instruction phase of the experiment. The experimenter read out these instructions aloud while simultaneously displaying the same text on presentation slides.*

Slide 1:

The money you earn for the environmental protection organization today will be donated to an organization that actively promotes climate protection. This means that greenhouse gas emissions are reduced directly and sustainably. To this end, fossil fuels are replaced by renewable energies, and energy-efficient technologies are implemented. The organization runs projects that enable, for example, the expansion of solar energy, hydro and wind power, biogas and biomass, as well as efficient stoves and other measures for direct greenhouse gas reduction.

Slide 2:

Every point you score in today's experiment for the environmental organization will be converted into money and donated to the organization. In this way, you are actively contributing to reducing CO₂ emissions immediately and directly."

...This might be the framing that some people see, but it ignores other motivations. Being vegetarian or taking the train does not have perceived as a sacrifice for the environment, but a desired option. It also suggests that reducing CO₂ emissions is a socially beneficial activity, while in some countries, like the USA, the federal government actively blocks citizens from doing the "right" thing. There might at least some reflection in the introduction on different aspects influencing pro-environmental behavior, and that it is not just a tradeoff between self and the environment.

We appreciate the reviewer's broader point that pro-environmental behavior is driven by many diverse motives and not always framed as a self-environment trade-off. We now wrote a paragraph focusing on this in the discussion. In particular, we now acknowledge that while our paradigm operationalizes pro-environmental action as a costly contribution to CO₂ reduction, real-world pro-environmental behaviors can be motivated by diverse considerations, including identity, values, and structural constraints, and are not always experienced as sacrifices.

Page 20-21. *"While our experimental paradigm focuses specifically on the cost-incurring dimension of pro-environmental action, we acknowledge that this represents only one facet of the broader motivational landscape. It is important to recognize that climate action is influenced by factors operating at multiple levels. At the individual level, cost-benefit evaluations play a role, but so do psychological and social drivers such as values, identity, or perceived norms^{26,57}. At the structural level, the availability of supportive infrastructure can facilitate or hinder pro-environmental choices, while at the national level, the broader context, such as the level of development of the country people live in or its fossil fuel reliance, can also influence people's willingness to act^{26,58-60}. Under certain conditions, these factors could also reduce or even eliminate the perception of costs, such that pro-environmental behaviors may be experienced as intrinsically desirable rather than as costly sacrifices. Importantly, our findings show that participants invested considerable effort and time on behalf of the environment in absolute terms: On average, they chose the costly option in 60.4% of trials in the effort task and 78.9% of trials in the time task. This demonstrates that, despite the relative inferiority of environmental rewards, participants were willing to incur personal costs for pro-environmental outcomes. Future research could, thus, build on these findings by using our paradigm to examine how individual, social, and structural*

factors interact with cost-incurring decisions, how subjective perceptions of costs (e.g., boredom) influence willingness to act, and how interventions might help reframe pro-environmental actions from perceived sacrifices into intrinsically valued behaviors.”

In general the experimental design is clever and well worked out, the analysis is sophisticated and the results are in line with my expectations. As mentioned above, the main challenge is understanding what we are actually measuring with the pro-environmental option.

We appreciate the reviewer’s recognition of the study’s strengths and fully agree that clarifying the pro-environmental option measures is critical. In our revision, we have taken several steps to address this.

We have now made what we mean by the pro-environmental option more explicit in the introduction.

Page 3. To address this, we conducted a preregistered experiment involving actual cost-incurring behaviours (physical effort and waiting times) with tangible outcomes: monetary rewards for the participant or for a CO₂ reduction program. By operationalizing pro-environmental motivation as the willingness to accept measurable personal costs (effort or time) to generate a concrete environmental benefit (CO₂ reduction), our paradigm moves beyond hypothetical scenarios or self-report measures to capture behavioral trade-offs in a controlled setting. Participants (n = 74) engaged in two novel tasks: an effort discounting task and a temporal discounting task. In both tasks, in half of the trials, participants could win money for themselves (“self” condition), whereas in the other half, they could win a monetary reward that would be donated to a CO₂ reduction program (“environment” condition).

We have also expanded the Methods section to clarify how the pro-environmental option was framed and communicated to participants.

Page 8. Participants were informed that every point they win in the CO₂ trials would be translated into money that would be donated to an organisation that actively combats climate change by reducing greenhouse gas emissions through projects that promote renewable energy sources, energy-efficient technologies, and sustainable solutions. The framing ensured participants were clearly informed that their choices represented contributions to greenhouse gas reductions, rather than broader prosocial causes (see SI, Section 2.5. for exact wording of the instructions).

More detailed comments:

The paper mentioned that in the environmental treatment a win is "translated into money that would be donated to an organisation that actively combats climate change by reducing greenhouse gas emissions through projects that promote renewable energy sources, energy-efficient technologies, and sustainable solutions". It would be helpful if information was provided what was communicated. At least it would be helpful to reflect on whether this is actually seen as a real contribution to reduce CO₂ emissions since it depends on the trust participants have on the impact of the investment. Giving money to an organization is not really a pro-environmental action, and donations are used a lot in business as green washing.

We thank the reviewer for this important comment. As mentioned above, we have now added detailed information about the exact wording that participants received regarding the environmental option. As the reviewer rightly points out, the extent to which participants perceive

such donations as genuine contributions to CO₂ reduction may depend on the trust they place in the implementing organization. Although we did not collect direct data on participants' level of trust, we were careful to communicate that we had selected a specific organization with a strong record of delivering on its promises and implementing effective greenhouse gas reduction measures. We also now acknowledge in the Discussion that variability in trust may influence how participants perceived the environmental option, and thus represents a potential limitation of our study.

Page 23. *"Lastly, although we carefully emphasized the strong track record of the selected organization, we cannot rule out the possibility that participants' willingness to choose the environmental option was influenced by their subjective perceptions of the organization, like e.g., the degree of trust they placed in it or the efficacy of measures suggested."*

Figure 1 is not clear. I would have expected a decision tree. It would have been appropriate if the action procedure was in the supplemental material.

We thank the reviewer for this suggestion. Figure 1 was now edited for clarity by, for example, adding labels to explicitly indicate what each option represents. In our paradigm, participants always made a binary choice between two options (costly vs. no-cost), and we designed Figure 1 to illustrate these options, with one example choice given per beneficiary (self, environment). In the main text we kept Figure 1 limited to one example per recipient (self vs. environment) to illustrate the decision process clearly without overloading the figure. To address the reviewer's point, we have now added to the Supplementary Information a decision tree for both the effort and the time tasks (SI, Figure S2), which we reference in the Methods section. For clarity, these trees are shown for one recipient only (self), since including both recipients would have doubled the complexity without adding substantive benefit. We also improved the action procedure figure to be more clear and opted for keeping it in the main text as it helps with understanding the multi-step procedure employed in the lab. The new supplemental figure can be seen below:

SI, Page 4. *"2.3. Task design decision tree"*

Figure 2. A. Decision tree for the effort task. On each trial, participants were given a choice between a no-effort option for a small reward and an effort option for a larger reward. If participants selected the effortful option, they were asked to exert the required force for at least 1 second within a 3-second window to earn the reward. If they did not exert sufficient effort they received 0 credits for the trial. Only self-trials selected for illustrative purposes. B. Decision tree of the time task. The task followed the same design as the effort task, with the only difference being that instead of exerting effort, participants had to endure a short waiting period. If they failed to endure the waiting period by skipping the waiting time after selecting it (analogous to failing to reach the required level of effort), they received 0 credits.”

Check the references both in the main text and in the supplemental material. Not all cited pubs are in the references.

Thank you for this, we carefully checked the citations and ensured they are all up-to-date.

Reviewer #3 (Remarks to the Author):

Review of the Manuscript

The paper presents a preregistered experimental study investigating how reward type and magnitude influence individuals' willingness to engage in tasks that vary in required effort and time. Specifically, the authors examine whether rewards targeted at the self versus the environment affect motivation, and they analyze discounting curves (reward vs. effort or time), linking estimated parameters to external variables such as climate change beliefs and policy support.

Overall, I am very positive about this paper for several reasons. It is well-written, clearly motivated, statistically well-powered, and preregistered (with accurate justifications in the case of deviations) Moreover, it introduces a potentially powerful behavioral paradigm for studying real-world pro-environmental behavior. The results are compelling—including the null findings, which are informative given the high statistical power. The paradigm is convincing in its ability to model trade-offs that resemble those encountered in everyday life. I particularly appreciate the attention given to potential differences between time and effort discounting. However, I do feel the paper would benefit from a dedicated section discussing limitations.

Disclaimer: I do not have extensive expertise in Bayesian statistics. While I understood the analyses as presented, I am not in a position to evaluate their technical rigor or completeness. My comments are intended to help improve the manuscript and/or inspire further reflection. I believe the paper is worthy of publication.

We thank the reviewer for the positive evaluation and the helpful suggestions!

Specific Comments and Suggestions

- **Policy Support Measure:** As this is a newly developed instrument, it may benefit from further elaboration. Given the study's focus on personal versus collective gain, it could be informative to explore—post hoc—whether items referencing personal versus societal costs differentially predict willingness to work or wait for personal versus environmental rewards. A factor analysis (FA) or principal component analysis (PCA), perhaps included in the appendix, could help clarify the internal structure of the measure.

We thank the reviewer for this suggestion. We agree that distinguishing between personal and societal costs would be an interesting angle for future work. However, the items in our policy support measure were not designed to make this contrast. While each policy was framed with an associated cost, we have no way of assessing whether these costs were affecting the participants personally. For example, policies increasing the price of fossil fuel vehicles or car-free cities would only be personal costs for those who drive. Similarly, while the regulation of coal use was described as leading to job losses in the coal industry, this would only represent a direct personal cost for participants working in that sector, whereas for most participants it would be perceived as a societal cost. Because of this heterogeneity, we do not think we can conduct a meaningful factor analysis contrasting “personal” versus “societal” costs with the current set of items.

Our team carefully considered whether conducting an exploratory factor analysis (EFA) would be appropriate instead. However, we decided against it due to the comparably small sample size of the present study. Methodological guidelines commonly recommend a minimum sample size of around 200 participants to ensure stable factor solutions (Gorsuch, 2014; Kline, 2014). Given our sample size ($n = 74$), we judged that an EFA would be underpowered and unlikely to provide meaningful or interpretable insights.

We have added to the limitations section that our measure does not allow us to determine whether the costs described in the items were perceived as personally relevant by participants, which may have influenced their responses. Future research could explicitly assess the personal versus societal relevance of policy costs and examine how this distinction relates to behavioral outcomes.

Page 23. *"Another important point is that our policy support measure does not allow us to determine whether the costs described in the items were perceived as personally relevant or primarily of societal relevance. Future research is needed to explicitly assess the personal versus societal relevance of policy costs and examine how this distinction shapes people's support."*

Gorsuch, R. L. (2014). *Factor analysis: Classic edition*. Routledge.

Kline, P. (2014). *An easy guide to factor analysis*. Routledge.

• **Line 194 ("participants were given no information"):** This phrasing is ambiguous. I assume it means that specific numerical information was withheld. However, was ordinal information provided, or were participants expected to infer levels through sensory cues? Including the exact instructions given to participants would help clarify this point.

We thank the reviewer for pointing out this ambiguity. Participants were sequentially presented with each of the five pie chart levels (one filled segment, two filled segments, etc.). After each presentation, they were asked to press hard enough to reach the required threshold indicated by a line on the screen. This procedure allowed them to experience directly what each level corresponded to and to gain a feeling for the relative difficulty. Thus, participants had clear ordinal information about the effort levels through the pie charts and practice trials, but they were not informed about the exact percentage of their own maximum force that each level represented. We rewrote the section to clarify that better.

Page 7. *"Participants experienced each effort level used later in the task (40, 50, 60, 70, and 80% of their MVC) to familiarize themselves with how hard they had to press to surpass the respective force thresholds. The effort levels were visually indicated as pie charts with 1-5 filled segments, corresponding to the five levels of effort. Participants were sequentially presented with each pie chart level and asked to press hard enough to reach the required threshold indicated by a line on the screen, allowing them to gain a feeling for the relative difficulty of each level. Importantly, they were not informed about the exact percentage of their own maximum voluntary contraction that each level represented."*

• **Time-Effort Calibration:** I found this aspect of the design intriguing. The supplementary information suggests it was challenging to implement. While the rationale is sound, it may be worth revisiting this in the discussion section to identify potential limitations. For instance,

differing discounting curves could inherently compromise the accuracy of indifference point identification. Conversely, the reliability of curve estimation may depend on the precision of those indifference points, which appeared problematic for some participants. This is not necessarily a flaw—your methodological care is commendable—but it would be helpful to acknowledge interpretive challenges and suggest avenues for future research. Many real-world activities (e.g., protesting) involve both time and effort.

We thank the reviewer for raising this important point. We agree that, given the different functional forms typically observed for temporal vs. effort discounting (hyperbolic vs. parabolic), any mapping based on a small number of indifference points will be at best an approximation. Thus, linear interpolation between the endpoints can deviate from the true (nonlinear) indifference curve at intermediate levels. We wrote a selection acknowledging this and suggesting future methodological improvements:

Page 23. “Relatedly, our time–effort calibration was designed to align the lowest and highest effort levels with corresponding waiting times, interpolating intermediate values. Given that effort and time follow different nonlinear discounting shapes, this approach is necessarily an approximation. Future work could increase precision by tracing multiple indifference points per participant (adaptive staircases) to estimate full indifference curves or by using conjoint/multi-attribute models that weight time and effort jointly.”

We agree with the reviewer that many real-world pro-environmental behaviors involve both time and effort costs simultaneously. Indeed, some existing paradigms, such as the Work for Environmental Protection Task (Lange & Dewitte, 2022), combine these dimensions without offering a way to disentangle their respective contributions. Our aim in the present study was to take the complementary approach: to design a paradigm that allows effort and time costs to be examined separately and directly compared. We believe this separation is valuable, as it enables more precise inferences about the mechanisms underlying each cost type, which can hopefully later inform research on situations where both costs occur together.

Moreover, is time perceived similarly across low and high effort conditions? Extremely low-effort tasks may be experienced as boring, which could influence discounting. Briefly flagging this as a topic for future exploration would be valuable.

We are not entirely sure to which aspect of the paradigm this pertains. To be clear, in the time task, low levels implied short waiting times, and high levels long(er) waiting times. If we understand the reviewer correctly, their concern would translate to short waiting times being more boring? Or does the question relate to whether effort is perceived similarly across low and high effort conditions, and the concern is that the low effort level might be too boring, and thus participants would avoid engaging? If the latter, the finding that the highest proportion of effortful option selected was for the lowest level of effort would speak against this.

If the comment more generally relates to subjective perceptions of the costs, note that we have collected data on this and report it in the Supplementary Information (SI, Section 3.6.). We collected subjective ratings for each level of the effort and time tasks, both before and after the experimental session. Specifically, for effort, participants rated each of the five levels on three items: physical demand (“How physically demanding was the task?”), effort (“How hard did you have to work to accomplish your level of performance?”), and unpleasantness (“How unpleasant

was it for you?”). For time, participants rated each level on three items: perceived duration (“How long did the passing of time seem to you?”), frustration (“How frustrating was it to wait for the time to pass?”), and boredom (“How boring was it to wait for the time to pass?”). We do see differences in subjective perceptions for all questions between the lowest and highest effort level, which is also in line with what our goals for these tasks (i.e., that people perceive these levels differently). In the case of boredom (which was measured only for the time task), we see that participants rated “boredom” of the lowest level of time demand quite low (post-testing average rating: 21 on a scale from 0 to 100) and higher on the highest level of the time demand (post-testing average rating: 61 on a scale from 0 to 100). Nevertheless, participants still chose to endure the longest waiting duration in over 70% of trials, suggesting that while boredom was present, it did not prevent them from selecting the costly option when they considered it worthwhile. While the focus of our work was more on the objective costs, we agree future work could build on this by investigating how subjective perceptions of costs (e.g., boredom, frustration) interact with objective time and effort demands in shaping pro-environmental decision-making. We now added this as a future direction in our discussion:

Page 21. *“Future research could, thus, build on these findings by using our paradigm to examine how individual, social, and structural factors interact with cost-incurring decisions, how subjective perceptions of costs (e.g., boredom) influence willingness to act, and how interventions might help reframe pro-environmental actions from perceived sacrifices into intrinsically valued behaviors.”*

• **Lines 601–602: I am not fully convinced that the literature on future-oriented time discounting—despite its robust evidence for hyperbolic discounting—can be directly applied to discounting in the present context. Why should current time discounting be analogous to future time discounting? It would strengthen the argument to include a line clarifying this assumption or to nuance the comparison.**

We thank the reviewer for raising this important point. To clarify, we see two complementary strands of research on temporal discounting. The first concerns experiential delay tasks, where individuals wait for seconds or minutes to receive an outcome (similar to our paradigm). The second is the classic intertemporal choice literature, which typically contrasts smaller–sooner versus larger–later monetary rewards (e.g., €5 now vs. €30 in six months). Although these contexts differ in timescale and framing, both literatures identify hyperbolic discounting as the best-fitting function (e.g., Coffey et al, 2003, Xu et al, 2016). While the longer time-scales (e.g., months) are more common in human research, the non-human animal literature studies discounting in the magnitude of seconds is very common and has also consistently shown hyperbolic discounting shape (Vanderveldt et al, 2016).

Coffey, S. F., Gudleski, G. D., Saladin, M. E., & Brady, K. T. (2003). Impulsivity and rapid discounting of delayed hypothetical rewards in cocaine-dependent individuals. *Experimental and Clinical Psychopharmacology*, 11(1), 18–25. <https://doi.org/10.1037/1064-1297.11.1.18>

Vanderveldt, A., Oliveira, L., & Green, L. (2016). Delay discounting: Pigeon, rat, human—does it matter?. *Journal of Experimental Psychology: Animal learning and cognition*, 42(2), 141.

Xu, P., González-Vallejo, C., & Vincent, B. T. (2020). Waiting in intertemporal choice tasks affects discounting and subjective time perception. *Journal of Experimental Psychology: General*, 149(12), 2289.

We added the following addition to the discussion on the computational modeling discussion:

Page 20. *“Although experiential delays in the range of seconds, as used in our paradigm,*

differ from the longer horizons more commonly studied in the temporal discounting research, both literatures converge on hyperbolic discounting as the characteristic functional form, suggesting continuity across timescales^{12,14,59,60}.”

• **Contingency Design Choice:** A notable feature of your design is the contingency applied when participants fail to follow through on their chosen plan—either by exerting insufficient effort or prematurely ending the waiting period. In such cases, they receive no reward. While this is methodologically defensible, one could argue that participants demonstrated good intentions and partial commitment, which might justify awarding them the default reward. From a fairness perspective, this could be seen as more balanced. Did you observe any participant reactions to this aspect of the design? Although not essential for this paper, a follow-up analysis could explore potential behavioral consequences. In real-world pro-environmental actions, individuals may attempt to act sustainably (e.g., cycling instead of driving) but encounter unforeseen obstacles (e.g., a flat tire or traffic jam). Such setbacks may lead to feelings of discouragement or perceptions of unappreciated effort. Your design may inadvertently simulate this dynamic. In effect, participants are choosing between a small, guaranteed reward and a larger, uncertain one—with a 90% chance of success or a 10% chance of receiving nothing. While this risk may be under their control in the case of waiting time, I question whether that holds true for effort-based tasks, especially at higher levels. It would be helpful to briefly justify this design choice and/or acknowledge it as a limitation or avenue for future research.

We appreciate this thoughtful comment. Our rationale for the “zero payoff if the costly option is not completed” contingency was to keep the paradigm incentive-compatible and preserve the intended trade-off between a small, guaranteed payoff and a larger, contingent payoff. If participants received the default reward even when not reaching the required threshold (or when aborting the wait), the costly option would dominate in expectation and the decision stage would no longer reflect a genuine willingness to incur costs; participants could simply select the costly option and then not follow through.

That said, “good intentions” did matter in our analyses: we did not exclude failed trials from the choice analyses, so intentions are fully represented in the choice data. In practice, such failures were rare: the success rate was 94.3% in the effort task and 99.6% in the time task. Regarding reactivity, we did not administer a targeted post-experiment question about this feature; however, participants had an open comment field and no one raised concerns about the contingency. We also tried to make the contingency fair and under participants’ control. For effort, thresholds were individually calibrated to each participant’s MVC, and success required holding above threshold for just ≥ 1 s within a 3 s window.

We agree that receiving no payoff after a failed attempt could, in principle, be discouraging. However, we believe there is no reason to assume that this discouragement would systematically differ between self- and environment-trials. Given that our main measure of interest is always the *relative difference* between these trial types, we think this design feature should not bias our key findings.

We rewrote parts of the methods section to justify our choice better.

Methods:

Page 8. “They had to attain the required effort level for (at least) 1 second during this

3-second window to attain the reward, which ensured that participants had a fair opportunity to succeed while also maintaining meaningful consequences for not reaching the target.”
“...This was implemented to keep the paradigm incentive-compatible and to preserve the intended trade-off between a small, guaranteed payoff and a larger, contingent payoff. Without this rule, participants could have selected the costly option without actually following through, and choices would no longer reflect a genuine willingness to incur costs.”

- **Line 311: Consider adding the term "reward" to clarify that the discounting pertains specifically to reward valuation. Is this interpretation correct?**

Thank you, we rewrote this sentence!

Page 10. *“To investigate the relationship between the difference in discounting of self and environmental rewards and support for climate policies, we used Kendall’s rank correlation coefficient.”*

- **Line 471: Please review this sentence. It appears that "individual difference" may have been used in place of "environmental reward" or a similar construct.**

Thank you for pointing out that this sentence is unclear. We rewrote it now:

Page 15. *“Additionally, the discounting was significantly stronger for environmental rewards compared to rewards for oneself (Wilcoxon signed-rank test: $V = 1009$, $p = 0.021$, one-tailed), with substantial interindividual variability, as indicated by Fig. 3D and by the standard deviation of the difference in discounting tendencies for self vs environmental outcomes which was around five times larger than the standard deviation of the discounting of self rewards (SD of $k_{diff} = 0.31$; SD of $k_{self} = 0.06$).”*

- **Line 340: The numeral "1" before the fraction seems out of place and may be a typographical error. If it does not convey meaningful information, consider removing it.**

Thank you for noticing this typo. We corrected it!

- **Line 484: The statement here reads as somewhat self-evident: if the standard deviation of variable 1 exceeds that of variable 2, then rank order reversals are expected. I believe the intended point is that, despite the overall main effect, a substantial subset of participants found the environmental reward more motivating. If so, rephrasing to emphasize this nuance would enhance clarity.**

Our intention with this comparison was to highlight the point that there was *much greater interindividual variability* in the difference between self- and environment-discounting (k_{diff}) than in discounting for self-outcomes alone. In other words, while the overall effect indicated stronger discounting for environmental rewards, this average conceals considerable heterogeneity: some participants showed the expected pattern of steeper environmental discounting, whereas others showed little difference or even found environmental rewards more motivating. We will rephrase this section to make this clearer. We added this clarification to the text:

Page 15. *“Additionally, the discounting was significantly stronger for environmental rewards compared to rewards for oneself (Wilcoxon signed-rank test: $V = 1009$, $p = 0.021$, one-tailed), with substantial interindividual variability, as indicated by Fig. 3D and the standard deviation of the difference in discounting tendencies for self vs environmental outcomes which was around 5 times larger than the standard deviation of the discounting of self rewards (SD of k diff = 0.31; SD of k self = 0.06). This indicates that while the average effect indicated stronger discounting for environmental rewards, this average conceals considerable heterogeneity: some participants showed the expected pattern of steeper environmental discounting, whereas others showed little difference or even found environmental rewards more motivating.”*

• **Concluding Note: As behavioral researchers, we often emphasize relative differences—such as the greater effectiveness of self-rewards over pro-environmental ones. However, it is striking that environmental rewards still elicit substantial effort and time investment in absolute terms. It would be worthwhile to highlight this somewhere in the manuscript, perhaps by quantifying the average amount of effort exerted or time spent “for the environment” per participant.**

Best of luck with the revision!

We thank the reviewer for this valuable suggestion. We agree that it is a great idea to highlight the absolute investment participants made in pro-environmental outcomes. We felt that quantifying the average amount of effort or time per participant would be less intuitive to interpret to the reader. Instead, we now report the proportion of trials in which participants chose the environmental option: on average 60.4% in the effort task and 78.9% in the time task. This addition underscores that, despite the relative inferiority of environmental rewards, participants nonetheless displayed a substantial willingness to incur costs for pro-environmental outcomes.

Page 21. *“Importantly, our findings show that participants invested considerable effort and time on behalf of the environment in absolute terms: on average, they chose the costly option in 60.4% of trials in the effort task and 78.9% of trials in the time task. This demonstrates that, despite the relative inferiority of environmental rewards, participants were willing to incur personal costs for pro-environmental outcomes. Future research could, thus, build on these findings by using our paradigm to examine how individual, social, and structural factors interact with cost-incurring decisions, how subjective perceptions of costs (e.g., boredom) influence willingness to act, and how interventions might help reframe pro-environmental actions from perceived sacrifices into intrinsically valued behaviors.”*